# Structural and biochemical evidence for the emergence of a calcium-regulated actin cytoskeleton prior to eukaryogenesis

Caner Akıl[1,2,3,7], Linh T. Tran [3,7], Magali Orhant-Prioux[4], Yohendran Baskaran [1], Yosuke Senju[3], Shuichi Takeda [3], Phatcharin Chotchuang[5], Duangkamon Muengsaen[5], Albert Schulte[5], Edward Manser[1,2], Laurent Blanchoin [4,6] & Robert C. Robinson [1,3,5 ✉]

Charting the emergence of eukaryotic traits is important for understanding the characteristics of organisms that contributed to eukaryogenesis. Asgard archaea and eukaryotes are the only organisms known to possess regulated actin cytoskeletons. Here, we determined that gelsolins (2DGels) from Lokiarchaeota (Loki) and Heimdallarchaeota (Heim) are capable of regulating eukaryotic actin dynamics in vitro and when expressed in eukaryotic cells. The actin filament severing and capping, and actin monomer sequestering, functionalities of 2DGels are strictly calcium controlled. We determined the X-ray structures of Heim and Loki 2DGels bound actin monomers. Each structure possesses common and distinct calcium-binding sites. Loki2DGel has an unusual WH2-like motif (LVDV) between its two gelsolin domains, in which the aspartic acid coordinates a calcium ion at the interface with actin. We conclude that the calcium-regulated actin cytoskeleton predates eukaryogenesis and emerged in the predecessors of the last common ancestor of Loki, Heim and Thorarchaeota.

[1] Institute of Molecular and Cell Biology, A*STAR (Agency for Science, Technology and Research), Biopolis, Singapore 138673, Singapore. [2] Department of Pharmacology, Yong Loo Lin School of Medicine, National University of Singapore, Singapore 117597, Singapore. [3] Research Institute for Interdisciplinary Science (RIIS), Okayama University, Okayama 700-8530, Japan. [4] CytomorphoLab, Biosciences & Biotechnology Institute of Grenoble, Laboratoire de Physiologie Cellulaire & Végétale, Université Grenoble-Alpes/CEA/CNRS/INRA, 38054 Grenoble, France. [5] School of Biomolecular Science and Engineering (BSE), Vidyasirimedhi Institute of Science and Technology (VISTEC), Rayong 21210, Thailand. [6] CytomorphoLab, Hôpital Saint Louis, Institut Universitaire d'Hématologie, UMRS1160, INSERM/AP-HP/Université Paris Diderot, 75010 Paris, France. [7] These authors contributed equally: Caner Akıl, Linh T. Tran. ✉email: rrobinson@okayama-u.ac.jp

The timing of the emergence of essential eukaryotic cellular functions is important for predicting the characteristics of organisms that gave rise to eukaryogenesis[1]. The regulated actin cytoskeleton is a protein machine that is integrated into many eukaryotic processes, such as cell movement and membrane remodeling[2]. Recently, a primitive regulated actin cytoskeleton has been demonstrated to exist at the protein level in Asgard archaea[3–7]. This includes homologs of actin[7,8], and its regulators, profilin[6,8] and cofilin/gelsolin[5], which control the assembly and disassembly of actin filaments, respectively. However, only one species of an Asgard archaea, the Lokiarchaeum *Candidatus* Prometheoarchaeum syntrophicum MK-D1 (MKD1), has been successfully isolated and cultivated[9]. Slow cell growth rates and low cell densities provide challenges to understanding the functions of protein machines in this isolate[9]. Previously, we used heterologous expression and in vitro characterization of Asgard profilins as a tractable route to understand aspects of the nature of non-isolated Asgard organisms[6]. Asgard profilins show activity in inhibiting spontaneous actin filament nucleation and in supporting filament elongation, characteristics shared with eukaryotic profilin. These observations were made in vitro using rabbit actin (rActin), testimony to the relatedness of the actin cytoskeletons in Asgard archaea and eukaryotes. Similarly, Asgard profilins, like eukaryotic profilins, lose their abilities to inhibit actin filament nucleation in the presence of phospholipids. However, some Asgard profilins do not bind to polyproline motifs[6] while others interact with modest affinities (~300 μM)[8]. In eukaryotes, more robust profilin/polyproline motif interactions (10–100 μM)[10,11] have been reported to recruit profilin:actin to filament nucleation machineries, which have yet to be discovered in Asgard archaea. Thus, Asgard archaea are predicted to have a functional actin polymerization machinery for which the nucleation is controlled by profilin and phospholipids, however, it is unknown if and how the force generated from actin polymerization is harnessed for a biological output.

A second aspect of actin dynamics in eukaryotes is the controlled depolymerization of actin filaments, which is highly regulated by the cofilin and gelsolin superfamilies of proteins[12–14]. In the Asgard phylum Thorarchaeota (Thor) distant homologs of cofilin and gelsolin exist, which in vitro can sever and cap rActin filaments and sequester rActin monomers[5], further adding weight to the relatedness of Asgard and eukaryotic cytoskeletons[7]. Cofilin and gelsolin domains share a similar core domain fold and likely a common ancestor[5,15]. Both proteins are capable of severing actin filaments and sequestering actin monomers, however gelsolin can also cap actin filaments[5,16,17]. Gelsolin is further distinguished from cofilin in that it is calcium controlled and it is comprised of six homologous domains in eukaryotes and mainly of two domains in Asgard archaea (Asgard 2DGel), whereas cofilin (ProGel in Asgard archaea) is a single domain protein[5,13,15,18]. There are two types of calcium-binding sites in gelsolins: Type I, between gelsolin and actin; and Type II, entirely coordinated by gelsolin residues[19]. Type I sites exist in human gelsolin domains 1 and 4, and in Thor2DGel domain 1, whereas Type II sites are present in all domains of human gelsolin and Thor2DGel[19–21]. Gelsolin appears to have arisen by two serial gene duplications of the core gelsolin domain, via a 2DGel-like sequence[5], followed by a further gene duplication of the three-domains to result in a six-domain protein in which pairs of gelsolin domains are most similar to each other: 1 and 4 (G1 and G4), 2 and 5 (G2 and G5), and 3 and 6 (G3 and G6)[5,18,22]. A record of the three-domain protein exists in many eukaryotic genomes and has similar functions to the six-domain protein[22]. One and two domain gelsolins are missing from the eukaryotic sequence databases[23], suggesting that these genes were not present in the last eukaryotic common ancestor (LECA), however,

calcium signaling to the actin cytoskeleton was present in LECA in the form of 3-6 domain gelsolins.

In summary, a functional actin has been demonstrated for Loki and Heimdallarchaeota (Heim)[7,8]; functional profilin for Loki, Heim, Thor and Odinarchaeota (Odin)[6,8]; whilst functional cofilin (ProGel) and gelsolin (2DGel) have been demonstrated only for Thor[5]. The potential existence of a calcium-controlled cytoskeleton in other Asgard phyla is unclear since the presence of homologous 2DGel sequences in databases is mosaic. Predicted 2DGels are currently absent from metagenome-assembled genomes (MAGs) for Odin, whilst Thor and Gerdarchaeota (Gerd) generally encode 2DGels comprising solely of two gelsolin domains, and Loki and Heim contain 2DGels with significant C-terminal extensions that are unrelated to gelsolin domains[7]. Currently, it is unresolved whether these extended 2DGels function in regulating actin filaments in a calcium-controlled manner. Here, we investigated the properties of extended 2DGels from Loki and Heim[3,4,9] through heterologous expression, X-ray crystallography, in vitro and heterologous *in cellulo* characterization. Calcium-regulated actin filament capping and severing, and G-actin sequestration, were observed for selected Loki and Heim 2DGels. This indicates that the last Loki, Heim and Thor common ancestor likely possessed a calcium-regulated 2DGel actin-filament depolymerization system, and that the emergence of calcium regulation of actin dynamics predates eukaryogenesis.

## Results

**Extended two-domain gelsolins**. BLAST sequence searches revealed that 2DGels exist in all Asgard phyla that are represented in the NCBI non-redundant protein sequences database, with the exception of Odinarchaeota[7]. Thor and Gerdarchaeota (Gerd) MAGs contain 2DGels without C-terminal extensions, whereas Helarchaeota (Hel) MAGs contain potential "2DGels" that contain a potential divergent gelsolin domain at the N-terminus[7]. Heim MAGs encode a number of hypothetical 2DGels of varied architectures. Examples include, Heim E29_bin46 which encodes a 208 amino acid (AA) 2DGel without a C-terminal extension (TET81546.1), similar in predicted domains to the characterized Thor2DGel (KXH72277.1, Fig. 1a)[5]. Whereas, Heim LC_3 (OLS25976.1, 284 AA) and B3_Heim (PWI47685, 409 AA) 2DGels have C-terminal extensions, beyond the two gelsolin-like domains, of 68 and 193 AAs, respectively. These C-terminal tails have no significant sequence homology to each other. In this study, we chose to investigate the Heim LC_3 284 AA 2DGel, which we refer to as Heim2DGel (Fig. 1a). There are a larger number of Loki hypothetical 2DGels in the sequence databases, partly due to the greater number of Loki MAGs. The most common 2DGel architecture, among Loki species, is typified by Loki GC14_75 (KKK44901.1, 335 AA), which we refer to as Loki2DGel (Fig. 1a). The C-terminal extension of Loki2DGel comprises a 69 AA linker (Link) followed by a potential zinc-binding domain (Zn) (Fig. 1a). The MKD1 genome encodes a similar architecture with a longer 79 AA linker, MKD1-2DGel (Fig. 1a). Both Loki2DGel and MKD1-2DGel were chosen to be investigated in this study.

Closer inspection of the AA sequences within the 2DGel core domains revealed that the signature calcium-binding residues in two calcium-binding sites (Types I and II) that are present in human gelsolin domain 1 (human G1) are conserved in both gelsolin-like domains (D1 and D2) from Thor, Heim and Loki (Fig. 1a, b). Furthermore, the signature actin-binding motif is present and located between D1 and D2, which is also found in WH2 proteins[24]. The four AA signature motif consists of a leucine, two basic residues, followed by threonine or valine. Thor and Heim contain classic motifs, LRRV and LKHV, respectively,

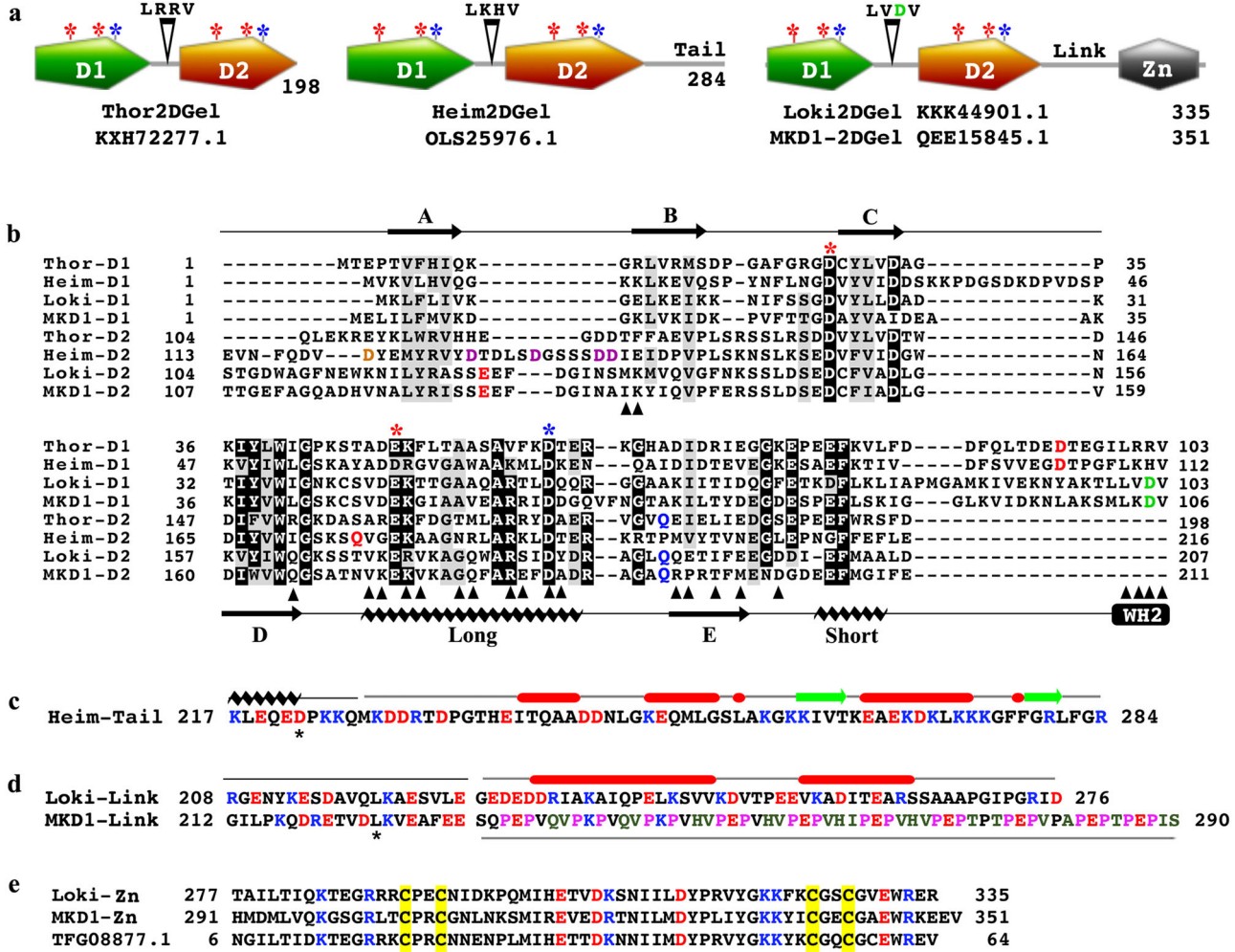

**Fig. 1 Sequence analysis of 2DGels. a** Schematic comparison of Thor2DGel with the extended Heim2DGel, Loki2DGel and MKD1-2DGel[5]. Asterisks indicate signature calcium-binding residues, Type I binding sites (blue) and Type II binding sites (red)[19]. The WH2-like motif appears between domain 1 (D1) and domain 2 (D2) and central "LKKT"-like 4 amino acids are indicated[24]. The Loki sequences have an unusual WH2-like motif "LVDV" in which the aspartic acid is highlighted in green. The C-terminal extended regions are referred to as Tail, Link and Zn. The protein sequence lengths are indicated. **b** Sequence alignment of the eight gelsolin domains from (**a**) based on the crystal structures determined in this manuscript. Secondary structure elements for Loki-D1 (Loki2DGel domain 1) are indicated above and below the alignment (black). Asterisks indicate signature calcium-binding residues, as in (**a**). Colored residues indicate additional calcium-binding sites. Black triangles indicate the actin-binding residues for Loki-D1. **c** Sequence of the Heim2DGel Tail. The first 10 residues are ordered in the crystal structure and include a helix and a calcium-binding residue (Asp222, black asterisk). The remaining residues are disordered in the crystal structure and their predicted secondary structure is indicated above, helices in red and strands in green[32]. Basic and acidic residues are colored blue and red. **d** Similar analysis for the linker regions of Loki2DGel and MKD1-2DGel. The first 20 residues are ordered and extended in the Loki2DGel/actin structure and include a mainchain calcium ion interaction (Leu220). Beyond this region the linker regions are not homologous. Loki2DGel is predicted to be helical, while MKD1-2DGel contains repetitive 6 amino acid repeat pink and green sequences that are predicted to be extended[32]. **e** The predicted $Zn^{2+}$-binding domain aligned with a 64 residue Loki protein. The $Zn^{2+}$-binding residues are highlighted in yellow and conserved charged residues are colored.

whereas Loki and MKD1 have an unusual motif (LVDV), in which the two basic residues are replaced by valine and aspartic acid (Fig. 1a, b).

The Heim2DGel Tail contains a high percentage (46%) of charged residues (13 acidic, 18 basic) with an estimated pI of 9.4 (Fig. 1c). Secondary structure prediction suggests that this extension may fold as a predominantly helical structure (Fig. 1c). The Loki2DGel Link between the gelsolin and Zn domains also contains many charged residues (38%, 17 acidic, 9 basic, pI 4.6) and secondary structure prediction suggests a helical structure (Fig. 1d). By contrast, the MKD1-2DGel Link is predicted to have an extended structure including nine copies of a pseudo six amino acid repeat "PEPVQV" (Fig. 1d), in which the prolines are unchanging and includes fewer charged residues (24%, 14 acidic,

9 basic, pI 4.7). The Loki and MKD1 Zn domains were modeled (Supplementary Fig. 1). The signature $Zn^{2+}$- binding residues (Cys-X-X-Cys)$_2$ are closely positioned, appropriate for potential zinc binding (Supplementary Fig. 1b). The Loki and MKD1 Zn domains have 34% charged residues (8 acidic, 12 basic, pI 8.8) and 31% charged residues (9 acidic, 10 basic, pI 7.8), respectively. Homologous Zn domain sequences are widely distributed in Loki sequences databases in potential 2DGel proteins and in unrelated multi-domain proteins, exclusively residing at the C-termini of these proteins. Furthermore, small proteins solely consisting of the Zn domain are also present in Loki (such as TFG08877.1, Fig. 1e). Besides the proposed Zn-binding motif, these sequences have a conserved extended loop with a tyrosine residue at its extremity (Supplementary Fig. 1). We were not able to

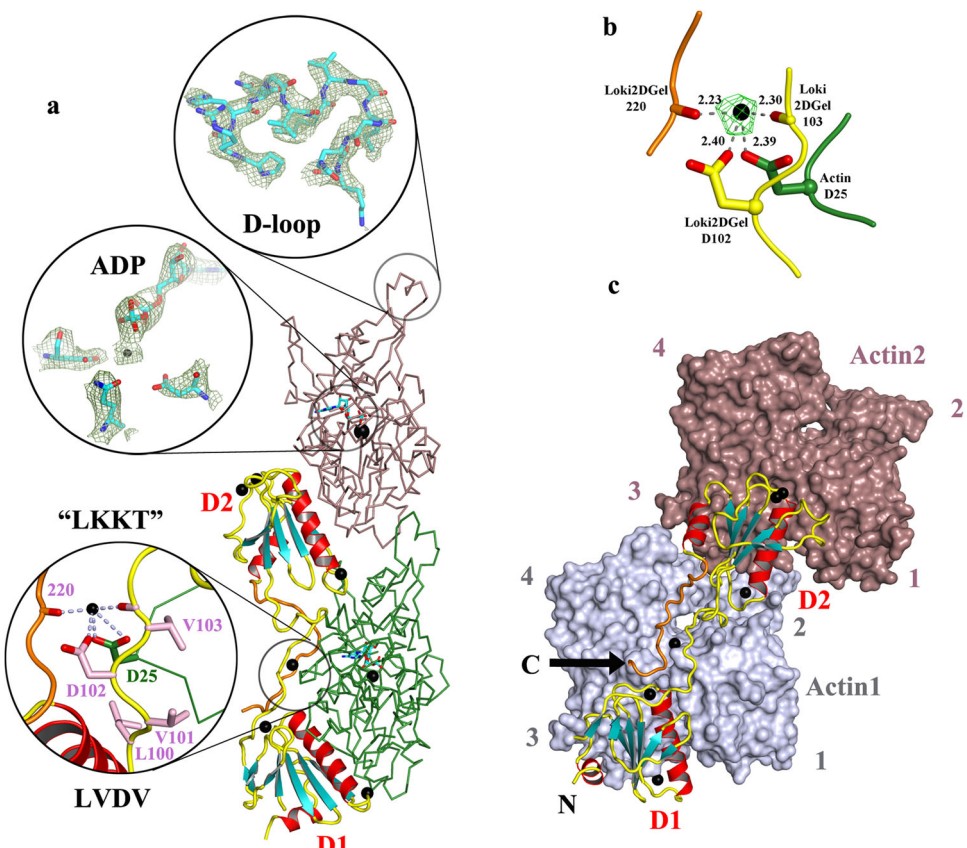

**Fig. 2 The Loki2DGel/rActin complex. a** Structure of Loki2DGel bound to two copies of rActin (side view). Loki2DGel shown as a cartoon with the ordered portion of the C-terminal extension in orange. Actins are depicted as ribbons (green and brown). The novel calcium-binding site that is coordinated by the Loki2DGel WH2-like motif[24], C-terminal extension and actin residue Asp25 is highlighted. The OMIT map electron density (contour level 1 σ) is shown around the D-loop and ADP are also highlighted. Calcium ions are depicted as black spheres. Data collection and refinement statistics for this and subsequent structures are found in Table 1. **b** The calcium-binding site in the WH2-like motif is shown with bonding distances to the calcium ion (Å). The Fo-Fc electron density (contour level 6 σ, peak 8.5 σ), before the placement of the calcium ion, is shown as a green mesh. **c** Front view of the Loki2DGel/actin complex. rActin is shown as surfaces (slate and brown) and Loki2DGel is shown in schematic representation. C and N indicate the C- and N-termini, respectively. Numbers refer to actin subdomains.

consistently detect homologous Zn domains in other organisms beyond Loki. Taken together, sequence analysis reveals that the C-terminal extensions are distinct between Loki and Heim phyla and show significant variability within each phyla.

**Structure of Loki2DGel.** To further investigate the similarities of Asgard gelsolins relative to each other and to their eukaryotic counterparts, we expressed and purified Heim-, Loki- and MKD1-2DGels. Loki2DGel crystallized in the presence of rActin and in the absence of other actin sequestering agents. The X-ray structure refined at 3.25 Å resolution revealed that Loki2DGel binds to two rActin monomers (Fig. 2 and Table 1). The two ADP-bound actin molecules are in a G-actin conformation and have ordered DNase I binding loops (D-loops, Fig. 2a), which include helical sections similar to the rhodamine-ADP-actin structure[25]. The gelsolin-like domains bind to the lower actin (Fig. 2) in an orientation that is similar to the first two domains of human gelsolin[21] (Supplementary Fig. 2). Following Loki2DGel domain 2, the subsequent 20 AAs of Loki2DGel Link region is ordered and lies on the surface of actin and extends back towards the first domain (orange, Fig. 2). This differs from human gelsolin in which G2 and G3 form a compact unit with G3 having a substantial interaction with actin[21] (Supplementary Fig. 2). Both post domain 2 architectures appear to stabilize the respective D2s

relative to the actin to which the D1s are bound (Fig. 2 and Supplementary Fig. 2). The remainder of the Loki2DGel Link and Zn regions have no interpretable electron density. The two calcium-binding sites observed in human gelsolin G1 are present and occupied in Loki2DGel D1 and D2 (Types I and II), and two additional calcium ions are observed (Fig. 2 and Supplementary Fig. 2). The Type I site in D1 coordinates rActin residue Glu167, which is conserved in Loki and MKD1 actins, whereas the Type I site in D2 does not coordinate an actin residue. One of the novel sites involves the central WH2-like "LKKT" motif of gelsolin, which is FKHV in human gelsolin, and LVDV (residues 100-103) in Loki2DGel (Fig. 1a, b)[24]. The aspartic acid in this motif coordinates a cation and the coordination sphere includes Asp25 from actin, which is conserved in Loki and MKD1 actins, and a backbone carboxyl from the post D2 Link region (residue 220). The Fo-Fc electron density map showed an 8.5 σ peak at this site, prior to the addition of the cation to the structure (Fig. 2b). The bonding distances of the coordination sphere atoms (2.23-2.40 Å) are appropriate for several types of cation from the crystallization conditions, $Ca^{2+}$, $Mg^{2+}$ or $Na^+$. We refined each of these cations individually in the structure. Only, $Ca^{2+}$ refined normally, characterized by a B-factor similar to the coordination sphere atoms and no unexplained Fo-Fc electron density (>3.0 σ). As such, the active conformation of Loki2DGel and its interaction with actin are stabilized via the calcium ion binding between the

**Table 1 X-ray diffraction data collection and refinement statistics.**

| | Heimdall2DGel/rActin (PDB code 7WHF) | Loki2DGel1/2rActin (PDB code 7WHG) |
|---|---|---|
| Data collection | | |
| Crystal | $P2_12_12_1$ | $P2_1$ |
| $a, b, c$ (Å) | 76.3, 100.4, 171.8 | 87.6, 72.1, 100.4 |
| $\alpha, \beta, \gamma$ (°) | 90.0, 90.0, 90.0 | 90.0, 92.1, 90.0 |
| Wavelength (Å) | 1.0 | 1.0 |
| Resolution (Å)[a] | 20.0-2.10 (2.14-2.10) | 20.0-3.25 (3.4-3.25) |
| $R_{merge}$ | 6.1 (65.1) | 16.5 (118.9) |
| $R_{meas}$ | 6.7 (71.4) | 18.3 (132.6) |
| $R_{pim}$ | 2.7 (28.6) | 7.7 (57.8) |
| $I/\sigma(I)$ | 24.7 (1.7) | 10.5 (1.4) |
| $CC_{1/2}$ | 0.954 (0.827) | 0.887 (0.523) |
| Completeness (%) | 95.3 (75.3) | 100.0 (99.8) |
| Redundancy | 5.9 (5.3) | 5.5 (5.1) |
| Refinement | | |
| Resolution (Å) | 20.0-2.10 (2.13-2.10) | 20-3.25 (3.46-3.25) |
| No. reflections | 63047 (1312) | 17387 (1504) |
| $R_{work} / R_{free}$ | 16.5/21.6 (21.7/29.3) | 17.5/23.2 (24.5/32.4) |
| No. atoms* | | |
| Protein | 5551 (A)/3597 (G) | 5799 (A)/1745 (G) |
| Ligand/ion | 74 (ATP, GOL)/18 ($Ca^{2+}$) | 54 (ADP)/9 ($Ca^{2+}$) |
| Water | 553 | 0 |
| B factors | | |
| Protein | 19.6/24.9 | 33.5/37.0 |
| Ligand/ion | 15.2/11.3 | 26.0/18.9 |
| Water | 33.6 | 36.9 |
| r.m.s deviations | | |
| Bond lengths (Å) | 0.013 | 0.011 |
| Bond angles (°) | 1.37 | 1.31 |
| Ramachandran Plot | | |
| Favored (%) | 98 | 96 |
| Outliers (%) | 0 | 1 |

WH2-like motif and actin. The net charge will change, on $Ca^{2+}$ binding, from the negatively charged Asp102 to the positively charged $Ca^{2+}$-bound Asp102, to mimic a basic residue. Thus, the LVDV motif is a calcium-regulated equivalent to the WH2-like LKKT motif. MKD1-2DGel did not crystallize in the presence of rActin. Sequence conservation with the ordered regions of Loki2DGel indicates MKD1-2DGel will interact with calcium and actin in a similar manner, including $Ca^{2+}$-binding in the LVDV motif (Fig. 1b) and actin interaction with the first 20 AA of the Tail (Fig. 1c). The actin-binding residues on Loki2DGel show a core conservation with MKD1-2DGel with a few residues of opposite character (Supplementary Fig. 1a). The actin sequences from Loki GC14_75 and MKD1 are identical in the Loki2DGel-binding site, and these residues show remarkable conservation with rActin (Supplementary Fig. 1c), in line with the complex formation between Asgard 2DGels and rActin[5].

**Structure of Heim2DGel**. We next determined the 2.1 Å resolution X-ray structure of Heim2DGel bound to a single actin in the absence of other sequestering agents (Fig. 3a–d and Table 1). D1 of Heiml2DGel binds in the classic site between actin subdomains 1 and 3, similar to Loki2DGel and human gelsolin (Fig. 2 and Supplementary Fig. 2). The chain continues, extending up the face of actin, with a central WH2-like motif (LRRT) before veering off to position D2 at a novel binding site on actin subdomain 4, which would not be expected to interact with a second actin in a filament. The C-terminal helix of D2 is extended, relative to Loki2DGel, ordering 10 residues of the Tail, beyond which there was no

interpretable electron density. Two orientations of Heim2DGel D2 are observed in the crystallographic asymmetric unit. The D2/actin binding site in copy 1 (Fig. 3a) is disrupted by the insertion of two glycerol molecules in the second copy (Fig. 3b), possibly due to stress from crystallographic packing. The two human gelsolin G1 calcium-binding sites (Type I and II) are present and occupied in each gelsolin domain of Heim2DGel. The Heim2DGel Type I in D1 coordinates rActin residue Glu167 in both copies of Heim2DGel, which is conserved in Heim actin. The Type I site in D2 coordinates rActin residue Glu241 in copy 1, which is conserved in Heim actin. Three additional calcium-binding sites were located Heim2DGel (Figs. 1a–c and 3a–c). Interestingly, several of these sites order an insertion region (Fig. 3e), relative to Loki2DGel D2, and this region forms an extensive interaction in the crystal packing that may suggest that the calcium ion stabilized loops form calcium-regulated protein:protein interactions. Comparison of the calcium-binding sites between Loki- and Heim2DGels (Figs. 1b and 3e) reveals the additional calcium-binding sites, beyond the Type I and II sites, are different in the two structures. Thus, the Heim2DGel structure reveals a novel binding site on actin for D2 that includes 10 residues of the Heim Tail, ordered insertions in D1 and D2, and unique calcium-binding sites.

**In vitro activities of Asgard 2DGels on actin dynamics**. Next, we investigated how the Asgard 2DGel interactions with actin translate into regulation of actin dynamics in vitro. In these experiments, we used eukaryotic actin since Asgard actins are difficult to produce on the scale needed for biochemical experiments. Interpretation of interactions of proteins from organisms that have diverged for ~2 billion years (Asgard archaea vs eukaryotes) has positive and negative aspects: negative, interactions may be weak or non-existence due to divergence in protein sequences, Asgard actins display ~50-60% identity to eukaryotic actins[6]; and positive, any interaction that has survived this period of divergence is likely to be essential and maintained through eukaryogenesis to extant eukaryotes. In the presence of calcium ions, MKD1-2DGel formed a complex with actin that eluted at around 160 kDa on a calibrated gel filtration column, whereas Heim2DGel eluted later at around 68 kDa (Fig. 4a–d). The positions of the elution peaks of the complexes were unaffected by varying the ratios between actin and the 2DGels, and SDS PAGE analysis suggested 1:2 and 1:1 ratios of MKD1-2DGel and Heim2DGel with actin, respectively (Fig. 4a–d and Supplementary Fig. 3), indicating that MKD1-2DGel may bind cooperatively to two actin subunits. These results are consistent with complexes observed in the crystal structures. By contrast, Loki2DGel did not sequester G-actin under the same conditions leading to viscous solutions not suitable for gel filtration experiments. We interpret this result to indicate that the interactions observed in the Loki2DGel/eukaryotic actin complex are weak due to the divergence, and are insufficient to prevent polymerization. Loki GC14_75 and MKD1 share 92% and 46% identity for actin and 2DGel, respectively (Supplementary Fig. 1a).

Subsequently, we employed pyrene-actin assays to monitor the effect of Asgard 2DGels on actin filament assembly and disassembly in the presence of calcium ions. The increase in pyrene fluorescence due to actin (5 μM) polymerization was not inhibited by the inclusion of Loki2DGel (10 μM) (Fig. 4e), whereas polymerization was inhibited by 2.5 μM and 5 μM MKD1-2DGel (Fig. 4f). The inhibition at a half molar ratio is consistent with the gel filtration data (Fig. 4a), which indicates a stable 1:2 complex of MKD1-2DGel sequestering the actin monomers. To assess the sequestering ability of the second gelsolin-like domain (D2) alone, a construct consisting of the isolated domain was engineered and purified (MKD1-D2).

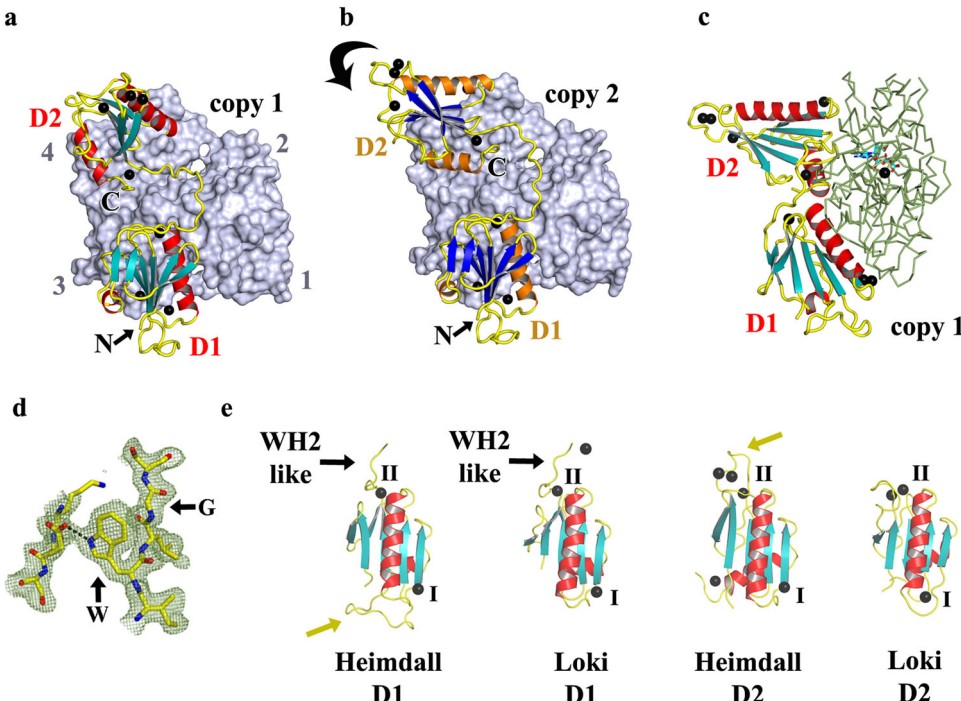

**Fig. 3 The structure of the Heim2DGel:rActin complex. a** Structure of one copy of the Heim2DGel/rActin complex from the crystallographic asymmetric unit. rActin is shown as a surface (slate) and Heim2DGel is shown in schematic representation. Calcium ions are depicted as black spheres and the N- and C-termini are labeled N and C, respectively. Actin subdomains are numbered. **b** Structure of the second copy of the Heim2DGel/rActin complex in which glycerol (not shown) has inserted into the D2/actin interface. The arrow indicates the movement of D2 relative to (**a**). **c** Side view of (**a**) with actin shown as a ribbon (green) for comparison with Fig. 2. **d** OMIT map electron density (contour level 1 σ) around the characteristic gelsolin WG motif in D1 copy 1. **e** Comparison of the individual gelsolin-like domains and their calcium-binding sites. I and II indicate the conserved calcium binding sites Type I and Type II, respectively. D1 and D2 refer to domains 1 and 2[19]. The C-terminal extension of D1 contains a WH2-like motif in each case[24]. Arrows (mustard) indicate insertions in the Heim2DGel domains.

MKD1-D2 (10 μM) was able to inhibit actin (5 μM) polymerization (Fig. 4g), indicative of sequestering activity of this domain. Finally, Heim2DGel was able to inhibit actin (5 μM) polymerization at a 1:1 ratio (Fig. 4h). In the absence of calcium ions, the Asgard 2DGels, including MKD1-D2, were unable to prevent actin polymerization (Supplementary Fig. 4a). Thus, actin monomer sequestration is $Ca^{2+}$ dependent for Heim and MKD1 2DGels.

In the depolymerization assay in the presence of calcium ions, polymerized actin (5 μM) was mixed with Asgard 2DGels before being diluted 100-fold, to below the critical concentration for actin polymerization. In this assay, Loki2DGel (1 μM) showed activity in slowing the depolymerization of actin as judged by the loss of pyrene fluorescence, consistent with filament capping, however, the capping activity was less potent than cytochalasin D at the same concentration[26] (Fig. 4i). By contrast, MKD1-2DGel was efficient at depolymerizing actin filaments at low concentration (5 nM, Fig. 4j), indicative of robust severing, followed by dissociation of actin subunits from the increased number of filament pointed ends. MKD1-D2 increased depolymerization at high concentrations (1-10 μM, Fig. 4k). These high concentrations suggest that MKD1-D2 is either able to sever filaments with low efficiency or can accelerate dissociation at filament ends. Lastly, Heim2DGel efficiently depolymerized actin filaments at low concentration (25 nM, Fig. 4l) indicating that the rate of severing by Heim2DGel is approximately 5-fold lower than MKD1-2DGel. The effects of Asgard 2DGels on actin depolymerization were not apparent in the absence of calcium ions (Supplementary Fig. 4b–e). These data indicate that actin filament severing and capping by Heim and MKD1 2DGels are calcium ion dependent.

To confirm the pyrene-actin depolymerization assays, we used high speed centrifugation to sediment actin filaments in the presence of Asgard 2DGels and analysed the composition of the soluble and pelleted fractions by SDS PAGE. Actin was found in the pellet in the absence of Asgard 2DGels, and in the presence of 2DGels but absence of $Ca^{2+}$, through inclusion of EGTA (Fig. 5a–d). Inclusion of $Ca^{2+}$ in the sedimentation experiments led to actin moving to the soluble fraction for MKD1-2DGel, MKD1-D2 and Heim2DGel but not for Loki2DGel (Fig. 5a–d). Hence, on the timescale of the experiment, these three constructs were able to disassemble the filaments. At low ratios of the 2DGel constructs to actin in $Ca^{2+}$, conditions under which substantial actin filaments were found in the pellet fractions, we did not observe high amounts of the 2DGel constructs in the pellets (Supplementary Fig. 5). Thus, these constructs do not decorate the sides of actin filaments. This is consistent with a mechanism of transient interaction with the sides of actin filaments for MKD1-2DGel and Heim2DGel followed by severing and capping, and possible dissociation of G-actin complexes. Furthermore, since MKD1-D2 does not decorate actin filaments (Supplementary Fig. 5), yet sequesters actin monomers (Fig. 4g), its role in MKD1-2DGel appears to be in biasing the actin conformation towards the G-actin form to accelerate barbed end dissociation following severing or end association by MKD1-2DGel, similar to the depolymerization mechanism proposed for eukaryotic twinfilin[27], a cofilin/gelsolin homolog. We found no evidence for bundling under the same conditions, judged by an absence of actin in the pellets of low speed sedimentation experiments (Supplementary Fig. 6a). Similarly, we saw no association of the Heim2DGel Tail or the Loki2DGel Zn domain with filamentous actin (Supplementary Fig. 6b).

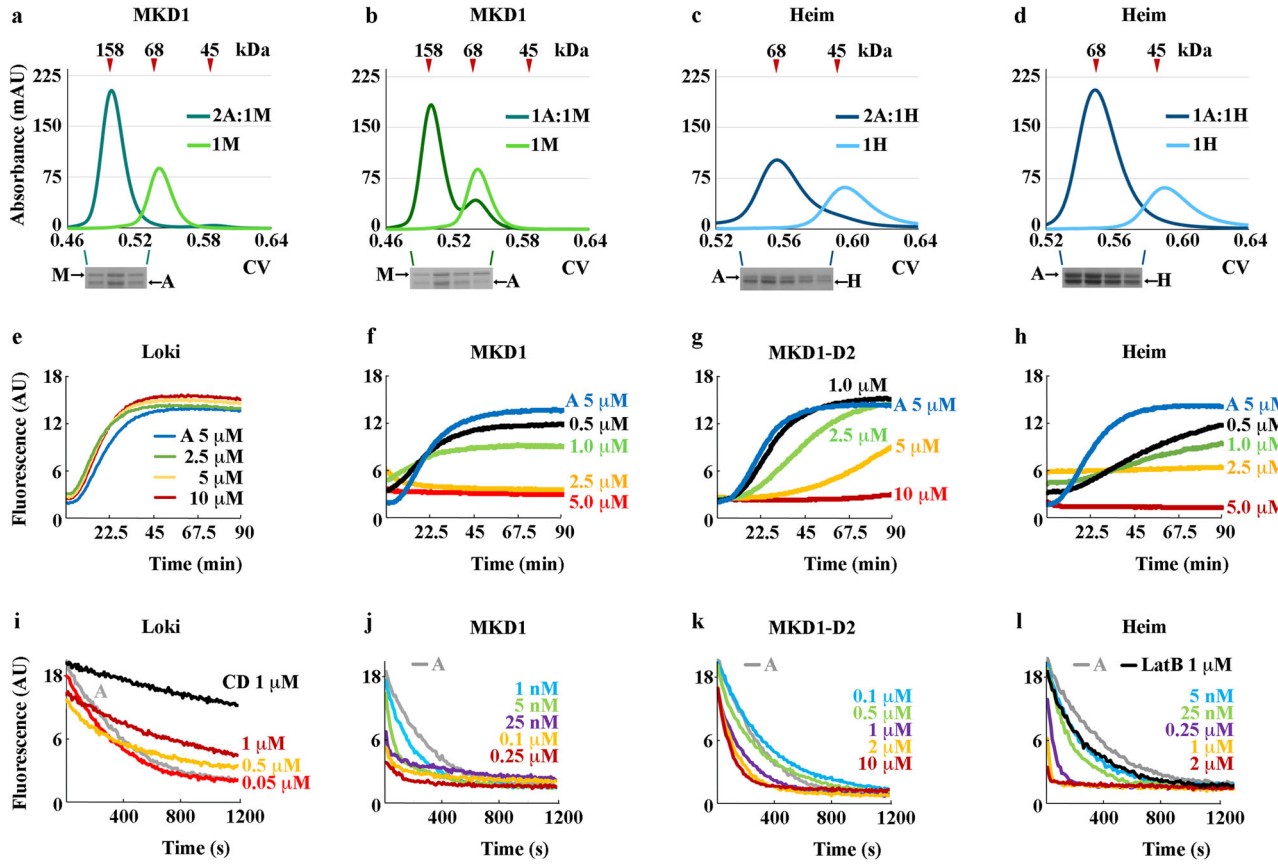

**Fig. 4 The activities of 2DGels with actin in Ca$^{2+}$. a–d** Gel filtration profiles of 2DGels alone or as complexes with actin. **a** MKD1-2DGel (1 M, light green) alone and at a 2:1 actin:MKD1-2DGel ratio (2 A:1 M, dark green). **b** 1:1 actin:MKD1-2DGel ratio (1 A:1 M, dark green). **c** Heim2DGel (1 H, light blue) alone and at a 2:1 actin:Heim2DGel ratio (2 A:1 H, dark blue). **d** 1:1 complex actin:Heim2DGel (1 A:1 H, dark blue). Molecular weight standard retention volumes are indicated by red triangles. Below SDS PAGE gels of the complex peaks. CV, column volume. **e–h** Pyrene-actin polymerization assays. 5 µM actin "A" was polymerized in the presence of increasing concentrations of 2DGels, color coded. **e** Loki2DGel, (**f**) MKD1-2DGel, (**g**) MKD1-2DGel domain 2 and (**h**) Heim2DGel. **i–l** Pyrene-actin depolymerization assays. 5 µM polymerized actin "A" was mixed with increasing concentrations of 2DGels, color coded, and diluted to 0.05 µM. **i** Loki2DGel, (**j**) MKD1-2DGel, (**k**) MKD1-2DGel domain 2 and (**l**) Heim2DGel. Fluorescence values were multiplied by 100 for comparison with (**e–h**). CD (cytochalasin D) and LatB (latrunculin B) indicate control depolymerization curves for actin filament capping and actin monomer sequestering molecules, respectively[26, 40]. Further gel filtration data and controls for the pyrene assays in EGTA are shown in Supplementary Figs. 3 and 4.

Finally, we used total internal reflection fluorescence (TIRF) microscopy to visually monitor actin filaments in the presence of Loki2DGel, MKD1-2DGel and Heim2DGel. In the presence of Loki2DGel and molecular crowding conditions, actin filaments annealed to form bundled filaments in EGTA, with less bundles being observed in the presence of calcium ions (Fig. 5e and Supplementary Movie 1). Thus, under these conditions Loki2D-Gel was able to change the behavior of actin, confirming an association between the two proteins as observed in the crystal structure and in the depolymerization data (Figs. 2 and 4i). By contrast, MKD1-2DGel quickly severed the filaments in calcium, but not in EGTA (Fig. 5f, g and Supplementary Movie 2 and 3). Heim2DGel did not show complete filament disassembly, however inclusion of Heim2DGel with polymerizing filaments changed the apparent rate of elongation of the actin filaments in a calcium dependent manner. In the presence of EGTA, and the absence of Ca$^{2+}$, the filaments elongated with an average apparent rate of ~5 subunits/sec, whilst in the presence of Ca$^{2+}$, the average apparent rate was close to zero (Fig. 5h, i and Supplementary Movie 4). The net halting of polymerization is likely a complex mix of actin monomer sequestration and filament severing and capping by Heim2DGel.

**In cellulo activities of Asgard 2DGels**. Previously we have shown that heterologously expressed Thor2Gel can disrupt actin filaments in response to calcium release in a human cell line[5]. Here we took the same approach in transfecting GFP-2DGel constructs in U2OS cells and observing the effect on actin filaments on releasing Ca$^{2+}$ from cellular internal stores using ionomycin (Fig. 6a). Under a fluorescence microscope, successful transfection was judged by GFP fluorescence and the presence of actin filaments via rhodamine fluorescence, after incubation with rhodamine phalloidin. In the absence of ionomycin treatment, GFP-positive cells resembled GFP-negative cells, indicating that the 2DGel proteins were inactive. After 10 minutes of incubation with ionomycin, GFP-Heim2DGel and GFP-MKD1-2DGel positive cells showed a substantial loss in rhodamine phalloidin stained F-actin, relative to the neighboring GFP negative cells (Fig. 6a), consistent with rapid dissociation due to severing. By contrast, the actin filament patterns for GFP-Loki2DGel positive cells appeared similar to the neighboring GFP-Loki2DGel negative cells (Fig. 6a), in line with the lack of depolymerizing activity observed in vitro (Fig. 4i). A GFP construct of MKD1-domain 2 alone (GFP-MKD1-D2) did not take apart the cellular actin filaments on ionomycin treatment, but appeared to co-localize

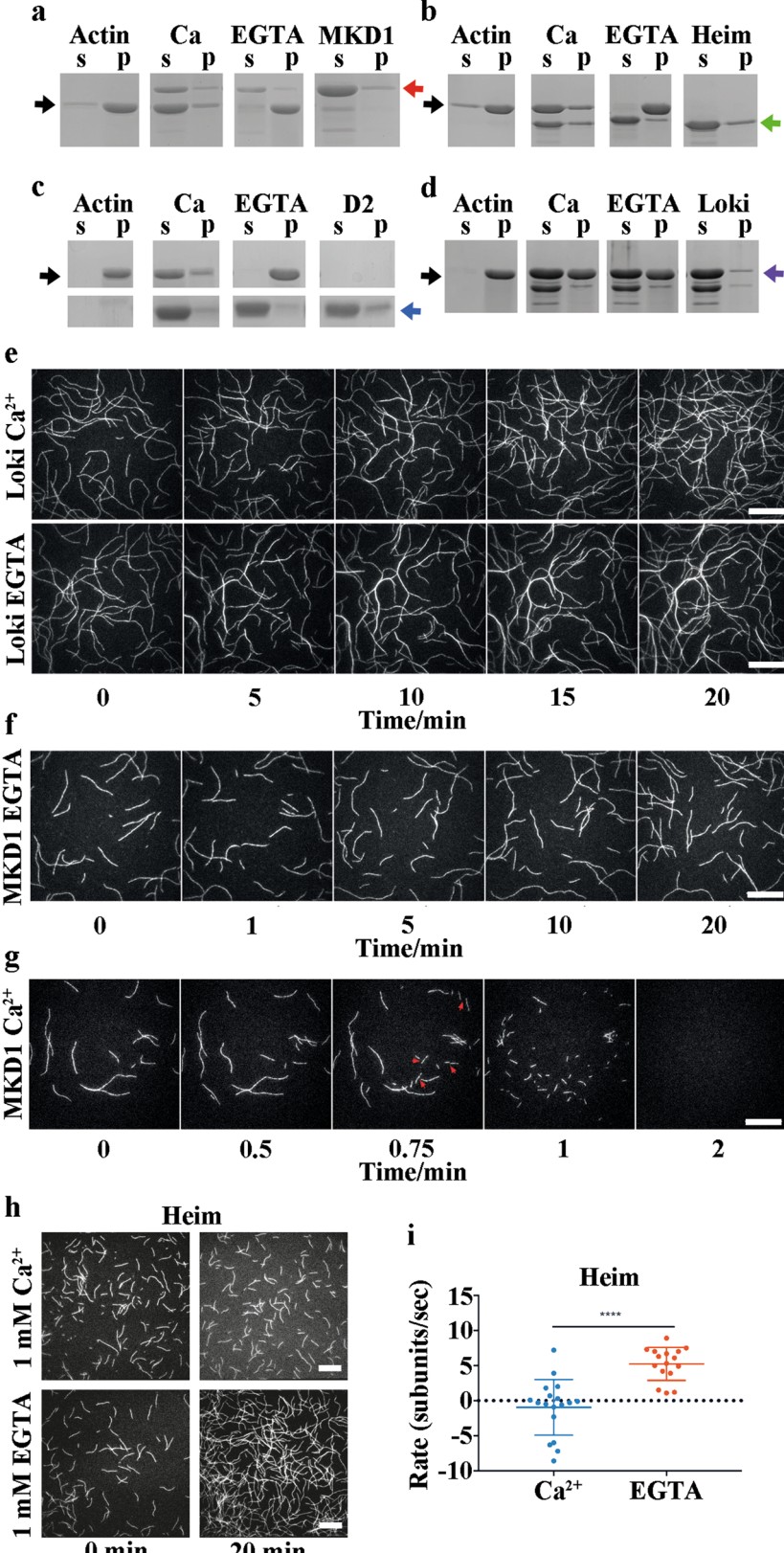

with the actin staining on calcium release (Fig. 6a). Thus, 2DGels from Heim, MKD1 and Thor are able to regulate actin disassembly in a eukaryotic cell line in response to $Ca^{2+}$ signaling[5].

In these transfection studies, we noticed that GFP-Heim2DGel often appeared to be excluded from the nucleus (Fig. 6b). We made a deletion construct (GFP-Heim truncated) lacking the

Heim2DGel Tail. Transfection of this construct resulted in a uniform GFP signal throughout the cell, including the nucleus (Fig. 6b). The loss of cellular actin filaments, on treatment with ionomycin, was comparable for the full length and truncated Heim2DGels (Fig. 6a and Supplementary Fig. 7), indicating that removal of the Heim Tail did not substantially effect its ability to

**Fig. 5 The activities of 2DGels on F-actin. a–d** SDS PAGE analysis of high speed sedimentation of F-actin (8 µM) treated with Asgard 2DGels. Soluble (s) and pellet (p) fractions are indicated. The position of actin migration is highlighted by black arrows. **a** MKD1-2DGel (MKD1, red arrow). Actin, actin alone; MKD1, MKD1-2DGel alone; Ca, 1:1 ratio in 1 mM Ca$^{2+}$; and EGTA, 1:1 ratio in 1 mM EGTA. The entire SDS PAGE gels are shown in Supplementary Fig. 4. **b** Heim2DGel (Heim, green arrow). **c** MKD1-2DGel domain 2 (D2, blue arrow). **d** Loki2DGel (Loki, purple arrow). **e** The effect of Loki2DGel on actin filaments. Actin was polymerized in 1 mM EGTA or 1 mM CaCl$_2$ for 10-20 mins. The time course follows the behavior of filaments on adding Loki2DGel (32 µM), followed by TIRF (Supplementary Movie 1). Similar time course for MKD1-2DGel (10 µM) in (**f**) 1 mM EGTA or (**g**) 1 mM CaCl$_2$. Arrows indicate severing. **h** Two time points for the assembly of F-actin (0.4 µM) in the presence of Heim2DGel (32 µM), in 1 mM Ca$^{2+}$ or 1 mM EGTA, imaged by TIRF microscopy (Supplementary Movie 3). **i** The apparent elongation rate of individual actin filaments from (**h**).

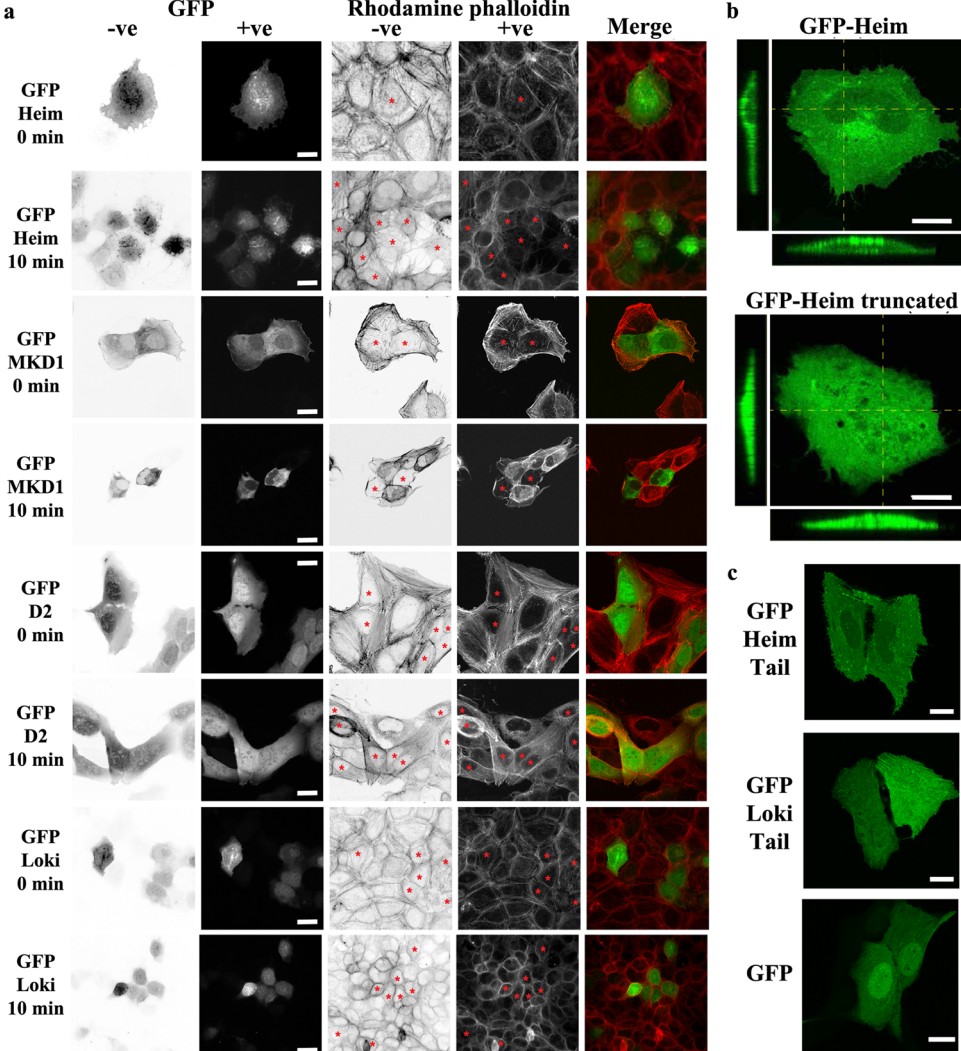

**Fig. 6 Localization and activities of heterologously expressed Asgard 2DGels in mammalian cells. a** U2OS cells expressing hybrid GFP constructs fused to Heim2DGel (Heim), MKD1-2DGel (MKD1), MKD1-2DGel domain 2 (D2), and Loki2DGel (Loki) were imaged before (0 min) or after (10 min) treatment with ionomycin to release cellular calcium stores. First two columns indicate the expression of the GFP-tagged proteins. "+ve" and "−ve" indicate normal and reversed images, respectively. The subsequent two columns detect F-actin structures imaged by rhodamine phalloidin staining. Red asterisks indicate cells expressing GFP-tagged proteins. The final column depicts merged images of the GFP (green) and F-actin (red) channels. Scale bars = 20 µm. **b** Typical GFP-channel images of U2OS cells expressing hybrid GFP constructs fused to Heim2DGel (Heim) or Heim2DGel without the Tail (Heim truncated) indicating that the presence of the Tail preferentially excludes the protein from the nucleus. Panels to the left and below show the Z-plane. Scale bars = 10 µm. **c** Typical GFP-channel images of U2OS cells expressing hybrid GFP constructs fused to the Heim2DGel Tail, Loki2DGel Tail or GFP alone. Scale bars = 20 µm.

dismantle actin filaments in response to Ca$^{2+}$. Finally, we transfected GFP fusion constructs of truncated Heim2DGel and Loki2DGel proteins without the gelsolin domains. These GFP-Tail constructs followed the localization patterns of the GFP-tagged native proteins. GFP-Heim-Tail was largely excluded from the nucleus, GFP-Loki-Tail was uniformly distributed throughout the cell, while the GFP control tended to accumulate in the nucleus (Fig. 6c and Supplementary Fig. 8).

## Discussion

In this report, we have demonstrated, in vitro and in a human cell line, that two-domain gelsolin proteins from Lokiarchaeota and

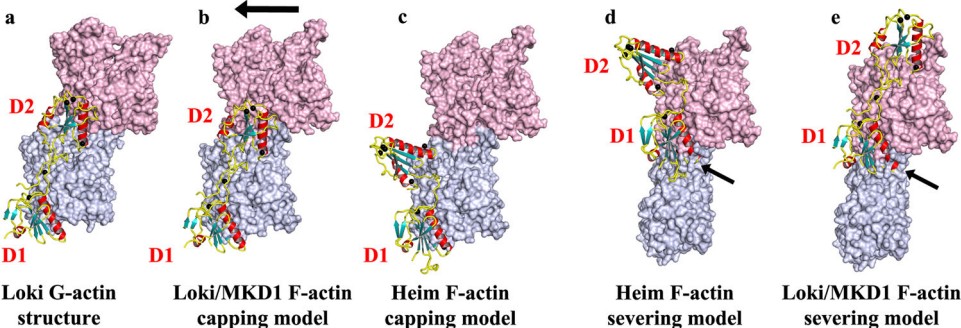

**Fig. 7 Models of 2DGel action. a** The structure of Loki2DGel bound to two molecules of G-actin. **b** Loki2DGel structure docked onto the lower actin (gray) an F-actin protofilament. The arrow indicates the translation of the second actin subunit in F-actin relative to the Loki2DGel/rActin structure. **c** Heimdall2DGel docked onto F-actin. **d, e** Models created by docking 2DGel onto the upper actin (pink) an F-actin protofilament. Arrows indicate steric clashes with the lower actin subunit. These proteins are incompatible with binding F-actin in these conformations. Steric clashes account for the sequestering and capping activities, via preventing a new actin monomer from joining another monomer or filament barbed end, respectively. Severing likely occurs when the 2DGel is located to the filament via D2 and/or the WH2 central motif allowing competition between D1 and the lower actin.

Heimdallarchaeota are capable of calcium-dependent eukaryotic actin filament regulation. We find evidence for actin monomer sequestration and actin filament severing and capping. These activities can be rationalized by inspection of the crystal structures in complex with rActin (Figs. 2 and 3). In the structures, Loki2DGel sequesters two subunits of G-actin and Heim2DGel a single G-actin. The position of the first gelsolin domain (D1), in both complexes, will prevent actin-actin interactions at this site, causing actin monomer sequestration. The mechanism of filament capping can be considered as equivalent to monomer sequestration with the D1's from each 2DGel preventing actin monomers from joining the barbed end of the filament (Fig. 7a–c).

For comparison, we calculated AlphaFold2 (AF2) predicted structures of Loki, Heim, MKD1 and Thor 2DGels bound to their respective Asgard actin (Supplementary Fig. 10). The lower actins, in each model, adopt an F-actin conformation and each D2 is located to the side of the two-actin modeled filament (Supplementary Fig. 10). We interpret these AF2 models to indicate the initial contact with actin filaments, where D2 and the WH2-like motif locate the 2DGels to the side of a filament allowing competition between D1 and the lower actin to induce severing (Fig. 7d, e). Interestingly, MKD1-2DGel has a two-residue insertion at the end of the D1 long helix (Fig. 1b), which will increase the steric clash with the lower actin (Fig. 7e, arrow), possibly providing an explanation for its superior filament severing activity. Thus, we propose that the filament severing and capping and monomer sequestration mechanisms are common for Heim, Loki, MKD1 and Thor 2DGels. The crystal structures of 2DGels with rActin (Figs. 2 and 3) likely represent post severing conformations of sequestered complexes, whereby D2 has dissociated from the second actin for Heim2DGel (Fig. 3), occupies a different site on actin for Thor2DGel[5], and the orientations of the two actins altered away from the filament conformation by Loki-2DGel (Fig. 7a, b). These different G-actin-bound complex conformations suggest variation in recycling of actin monomers in the different phyla.

Loki2DGel shows little biochemical activity with regards to actin monomers or filaments, while MKD1-2DGel is robust in severing filaments and in sequestering actin monomers. However, Loki2DGel was capable in sequestering rActin in the crystallization experiment. Sequence homology, between Loki2DGel and MKD1-2DGel, suggests that they will interact with actin through the same surfaces (Fig. 1b and Supplementary Fig. 1a, c). AF2 predicted structures of Loki and MKD1 2DGels bound to their respective Asgard actins also support a similar binding mode for Loki and MKD1 2DGels (Supplementary Fig. 10). We attribute the minimal biochemical activity for Loki2DGel to be due to

divergence in the sequences, resulting in a lower affinity of Loki2DGel for actin relative to MKD1-2DGel, which is insufficient to compete with the affinities of rActin:rActin interactions in standard actin assembly buffers. Inspection of the numbers of gelsolin-like proteins in both species of Loki is revealing. The fully sequenced MKD1 genome has a single 2DGel, two ProGels and two 1DGelXs[5,7]. By contrast, the incomplete GC14_75 MAG has two 2DGels (Supplementary Fig. 1a), four ProGels and three 1DGelXs. Thus, the differences in sequence that result in differences in the activities of Loki and MKD1 2DGels may represent functional diversification through paralogization in these species, and/or an in vitro variance in modulating the non-native rActin. The MKD1-D2 construct was competent in sequestering rActin monomers (Fig. 4g), indicating that D2 may impose a G-actin conformation on the bound subunit preventing it from undergoing a G-to-F actin transition in order to polymerize. D2 also disrupts the rActin:rActin contacts away from a filament orientation (Fig. 7a, b). This single gelsolin domain construct provides a model for the ancestral calcium-controlled gelsolin domain, prior to gene duplication, since its actin activities are regulated by calcium (Supplementary Fig. 9).

There are similarities and differences between the calcium-binding sites between the Asgard phyla (Figs. 1b and 3e). The structure of Thor2DGel bound to a single actin monomer showed occupancy of Type I and II sites in D1 but only the Type II site in D2, despite conservation in the signature binding residue in the D2 Type I site (Fig. 1b)[5]. In Heim2DGel and Loki2DGel all four sites are occupied (Fig. 3e). The Type I sites mediate interaction between gelsolin and actin[5,28]. Thus, we propose that the D2 calcium Type I site plays a regulatory role in locating D2 to the side of an actin filament and this calcium ion may dissociate from the Thor2DGel/G-actin complex. Beyond these common calcium-binding sites Loki2DGel and Heim2DGel contain two and three unique calcium-binding sites, respectively, that are not present in Thor2DGel (Fig. 3e)[5]. Thus, calcium-control of 2DGels shows variation between phyla.

The C-terminal extensions of Loki2DGel, MKD1-2DGel and Heim2DGel are distinct, while Thor2DGel contains no extension (Fig. 1). We have determined that the first 10-20 residues of these extensions are important for the actin interaction and contribute to unique calcium-binding sites (Figs. 1c, d, 2 and 3). Beyond these regions the roles of the Tail regions are unclear. The Tail regions contain many charged residues, possibly suggesting further calcium regulation. The differential nuclear exclusion of Heim2DGel, in eukaryotic cells (Fig. 6b, c), may hint at a role in directing cellular location. Furthermore, the Loki2DGel and

MKD1-2DGel Zn domains have a conserved extended loop with a tyrosine residue (Supplementary Fig. 1), and a homologous sequence is predicted to be discrete Zn domain protein (Fig. 1e). This conserved feature likely has binding potential to a cellular feature or protein complex. We hypothesize that these distinct 2DGel Tails will produce variation in the actin regulation in the different Asgard phyla.

In summary, calcium-controlled actin filament disassembly is present within the 2DGel proteins from Loki, Heim and Thor Asgard phyla. This indicates that their last common ancestor will have possessed a 2DGel which responded to calcium ions. Sequence analysis has identified several potential eukaryotic-like calcium channels in Asgard archaea genomes, which have been predicted to control calcium influx into these organisms[29]. In particular, Loki, Heim and Thor contain potential Calcium Load-activation Calcium Channels (CLAC), and Loki and Heim potential presenilins[29]. Thus, Asgard archaea may use calcium channels to regulate calcium levels, and in turn, control actin dynamics. Variations in 2DGel calcium-binding sites, positioning of D2s relative to actin, and distinct C-terminal Tails for Loki, Heim and Thor indicate significant evolution of these proteins since their common ancestor. We speculate that the ancestral Asgard actin cytoskeleton was immature in the common ancestor, and that the system further evolved in each lineage, resulting in differences in calcium control and actin disassembly to suit different environments and organism lifestyles. Nonetheless, the last common ancestor of Loki, Heim and Thor, and possibly the last Asgard archaea common ancestor, will have possessed an actin cytoskeleton under the control of calcium signaling, and these features predate eukaryogenesis.

## Methods

**Proteins**. Protein domain homolog identification was carried in BLAST[30] using reported Asgard sequences[3–5]. Default parameters were used in the online BLASTP interface (https://blast.ncbi.nlm.nih.gov/) in searching the non-redundant sequence database (2022/05/21). Potential calcium-binding gelsolin domains were identified by inspection of pairwise sequence alignments to locate the signature "WxG" gelsolin motif and the calcium-binding residues. The searches were not limited by E-value cut-offs. Domain architectures of proteins were created in Prosite MyDomains[31]. Secondary structure prediction was carried out in JPred4[32] and model building in AlphaFold2[33] implemented in ColabFold[34] or as an in-house installation. The Asgard 2DGel sequences were synthesized and codon optimized for *E. coli* (GenScript) and proteins were expressed and purified as described previously[5]. Truncation constructs comprised the following residues, MKD1-D2 (103-228), Heim2DGel Tail (227-270) and Loki2DGel Zn (273-335). Actin was purified and pyrene-labeled from acetone powder extracted from rabbit or chicken skeletal muscle[35].

**Crystallization**. Crystals of Heim2DGel/rActin and Loki2DGel/2rActin complexes were grown from 1:1 molar ratios (400 μM Asgard gelsolin: 400 μM rabbit actin) of the proteins were under the following conditions: Heimdall2DGel/rActin, 0.1 M HEPES, pH 7.0, 10% w/v polyethylene glycol 6000, at 297 K; Loki2DGel/2rActin, 0.1 M Tris-HCl, pH 7.0, 0.2 M magnesium chloride hexahydrate, 10% w/v polyethylene glycol 8000, at 291 K. CaCl₂ was added, to a final concentration of 1 mM, to the Asgard 2DGel/rActin mixtures prior to crystallization. All crystallization trials were performed using the sitting-drop or hanging-drop vapor-diffusion methods at 1:1 ratios of protein mixture to precipitate. Crystals were flash frozen in the crystallization buffer, which was supplemented by 25% glycerol prior to X-ray data collection.

**Structure determination, model building and refinement**. Native X-ray diffraction datasets from single crystals of Heim2DGel/rActin; and Loki2DGel/2rActin were collected on a RAYONIX MX-300 HS CCD detector on beamline TPS 05 A (NSRRC, Taiwan, ROC) controlled by BLU-ICE (version 5.1) at λ = 1.0 Å. Data were indexed, scaled, and merged in HKL2000 (version 715)[36] (Supplementary Table 1). Molecular replacement using the Heim2DGel/rActin; and Loki2DGel/2rActin datasets using the native actin (PDB code 3HBT)[35] was carried out in the PHENIX suite (version 1.13-2998) Phaser[37]. The model of Heim2DGel/rActin was extended in AutoBuild[37]. All manual adjustments to the models and refinement were carried out in Coot (version 0.8.9 EL)[38]. All final models were verified for good stereochemistry in PHENIX suite (version 1.13-2998)[37] MolProbity[39] (Table 1). The final Heim2DGel/rActin model has 2 copies in the asymmetric unit consisting of Heim2DGel residues 1-226 (with no observable electron density for the last 58 residues) for chain G, while chain C ends at residue 225. The two copies of rActin comprise chain A, residues 5-37, 51-61 and 65-375, and chain B, 5-35, 53-60 and 65-375. A combined 16 calcium ions are

associated with the Heim2DGel chains and one each to the two actins. Two molecules of glycerol are sandwiched between Heim2DGel chain C and actin chain B. The final Loki2DGel/2rActin model consists of Loki2DGel residues 2-84, and 89-226 (with no observable electron density for the last 109 residues) and two copies of rActin bound to ADP (chain A, residues 2-372; chain B, 5-375) with ordered DNase I binding loops. 7 calcium ions are associated with Loki2DGel and one each to the two actins. The lack of density for the C-terminal extension of Heim2DGel and Loki2DGel may be due to flexibility in these regions or to proteolytic cleavage in the crystallization drops. Protein structures were illustrated in PyMol (http://www.pymol.org).

**Pyrene actin assays**. For the pyrene-actin assembly assay, 5 μM of chicken actin (5% pyrene labeled) in buffer A (2 mM Tris-HCl, pH 7.5, 0.2 mM ATP, 0.2 mM CaCl₂, 1 mM DTT) was incubated with a 20-fold dilution of 20X Mg-exchange buffer (1 mM MgCl₂, 3 mM EGTA) for 2 min to exchange the calcium ion for magnesium. The actin was mixed with Asgard 2DGel proteins in the presence of 1 mM CaCl₂ or EGTA (final concentration), then 10X KMI buffer (500 mM KCl, 10 mM MgCl₂, 100 mM imidazole-HCl, pH 7.0) was added to initiate polymerization (total volume 100 μl). For pyrene-actin disassembly, chicken actin (5 μM; 20% pyrene labeled) was polymerized by the addition of 10X KMI buffer for 120 min at room temperature. 2 μl of pyrene-F-actin was diluted with 198 μl of the solution containing Asgard 2DGel proteins in buffer A with 1 mM CaCl₂ or 1 mM EGTA. Pyrene fluorescent was monitored at an excitation at 365 nm and emission 407 nm in a fluorescence spectrophotometer (Infinite® 200 PRO, Tecan).

**Sedimentation assay**. Asgard 2DGels and chicken F-actin (8 μM) were incubated in 1X KMI buffer with 1 mM CaCl₂ or 1 mM EGTA for 2 h at room temperature. The samples were centrifuged for 30 min, 20 °C at 10,000 × g or 150,000 × g. The supernatant and pellet were separated and analyzed by SDS-PAGE.

**Size exclusion chromatography**. In total, 20 μM of chicken G-actin was mixed with Heim2DGel or MKD1-2DGel with different ratios (1:1), (1:2) and (2:1) in 1X KMI-Ca buffer (50 mM KCl, 10 mM imidazole, 1 mM MgCl₂, 1 mM CaCl₂). After incubating 30 min, the samples were loaded on a gel filtration column (Enrich SEC 650, Bio-Rad), equilibrated with the same buffer. The peak fractions monitored by absorption at 280 nm were analysed by SDS-PAGE.

**TIRF assays**. Total Internal Reflection Fluorescence (TIRF) assays were carried out as previously described[5]. Briefly, actin assembly was initiated by addition of the actin polymerization mix (2.6 mM ATP, 26.7 mM DTT, 50 mM KCl, 5 mM MgCl₂, 10 mM Hepes, pH 7.5, 3 mg/mL glucose, 20 μg/mL catalase, 100 μg/mL glucose oxidase, 0.2% wt/vol bovine serum albumin [BSA] and 0.25% wt/vol methylcellulose) either supplemented with 1 mM EGTA or 1 mM CaCl₂, containing actin monomers (0.4 μM, 20% Alexa488-labeled) and 32 μM Heim2DGel. The effect of Loki2DGel and MKD1-2DGel on actin filaments (0.6 μM, 12% Alexa488-labeled, polymerized for 10-20 mins) was followed after adding 32 μM Loki2DGel or 10 μM MKD1-2DGel. Actin filaments were visualized with a Total Internal Reflection Fluorescence (TIRF) microscope (Roper Scientific) equipped with an iLasPulsed system and Evolve EMCCD camera (Photometrics) using a 60x or 100x Olympus APO TIRF oil-immersion objective. Images were acquired every 15 s using MetaMorph software (Universal Imaging).

**U2OS localization and cellular calcium release experiments**. U2OS cells were cultured in high glucose Dulbecco's modified Eagle's (DME) media with 4500 mg/L glucose supplemented with 10% Fetal Bovine Serum (FBS) (Hyclone) or in McCoy's 5 A medium (Thermo Fisher Scientific) supplemented with L-glutamine and 10% fetal bovine serum (FBS, Nichirei) and maintained at 37 °C with 5% CO₂. Mycoplasma contamination in cell cultures was routinely tested using the PCR mycoplasma detection set (Takara Bio). The EGFP-tagged gelsolin expression plasmids were transfected into cells using the Xfect transfection reagent (Takara Bio). U2OS cells were seeded onto 22 × 22 mm glass coverslips in 35 mm culture dishes and grown to sub-confluence. To induce intracellular calcium mobilization, the cell culture media was replaced with fresh media containing 1% FBS and 10 μM ionomycin (LKT Labs, Inc), and incubated for 10 min before fixation[5]. After 24 h incubation, cells were fixed with 4% paraformaldehyde (Nacalai Tesque, Inc.) in PBS for 15 min at room temperature, and permeabilized with 0.1% TritonX-100 in PBS for 5 min. To reduce nonspecific background staining, cells were pre-incubated with PBS containing 1% BSA for 30 min, then F-actin was stained with Alexa Fluor 647 Phalloidin (Thermo Fisher Scientific) in PBS containing 1% BSA for 30 min. The samples were mounted with Fluoro-KEEPER antifade reagent with DAPI (Nacalai Tesque, Inc.) or with fluorescent mounting medium (Thermo Scientific), and observed under an FV1200 confocal laser scanning microscope (Olympus) or using a Zeiss Axioplan2 microscope equipped with CoolSnap HQ cold CCD camera at 63X magnification. Image analyses were performed using ImageJ (NIH, USA).

**Statistics and reproducibility**. Pyrene-actin biochemical experiments were repeated 3 times with similar results (Supplementary Fig. 11).

**Reporting summary**. Further information on research design is available in the Nature Research Reporting Summary linked to this article.

## Data availability

The atomic coordinates and structure factors have been deposited in the Protein Data Bank under the accession codes 7WHF and 7WHG.

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

## Acknowledgements

We thank Professor Jian-Ren Shen and Professor Yuichiro Takahashi for use of reagents and access to equipment, and Hiroyuki Imachi and Masaru Nobu for sharing MK-D1 sequence data. We thank Yoshihito Kitaoku for implementing the in-house version of AF2. This work was supported by JST CREST, grant number JPMJCR19S5, Japan; by the Japan Society for the Promotion of Science (JSPS), grant number JP20H00476 and 20K06589 (YS), by the Moore-Simons Project on the Origin of the Eukaryotic Cell, grant number GBMF9743; by A*STAR, Singapore National Medical Research Council (NMRC grant OFIRG/0067/2018); by Vidyasirimedhi Institute of Science and Technology (VISTEC); and by Wesco Scientific Promotion Foundation (YS). We thank the experimental facility and the technical services provided by the Synchrotron Radiation Protein Crystallography Facility of the National Core Facility Program for Biotechnology, Ministry of Science and Technology and the National Synchrotron Radiation Research Center, a national user facility supported by the Ministry of Science and Technology, Taiwan, ROC.

## Author contributions

C. A., L. T. T., Y. B., L.B., Y. S., S. T. and R.C.R. conceptualised and designed research. C. A., L. T. T., M. O.-P., Y. S. and R. C. R. performed research. E. M., L.B. and R. C. R. coordinated data analyses with contributions from C. A., L. T. T., Y. B., M. O.-P., Y. S., S. T., P. C., D. M. and A. S. R. C. R. wrote original draft of the manuscript with significant contributions from C. A., L. T. T., Y. B., M. O.-P. and Y. S. All authors read and approved the final manuscript.

## Competing interests

The authors declare no competing interests.
