## [Peer Review File · Communications Biology]

Reviewers' comments:

Reviewer #1 (Remarks to the Author):

The manuscript is very well written and you can see that the authors have studied the subject extensively. The cross-linking to Norse mythology is very nice. However, the text does not address the effects of pH and temperature on gelsolin and its ability to bind calcium. Nevertheless, the paper has added value for research on gelsolin.

Reviewer #2 (Remarks to the Author):

In their contribution, Akil and coauthors show that the gelsolin homologs in two Asgard archaea regulate eukaryotic actin, both in vitro and in eukaryotic cells, and that calcium controls the action of gelsolin. The authors conclude that calcium regulation of actin predates the last common ancestor of Asgard (LCAs).

This review is limited to the protein alignment and searching methods and the evolutionary perspective of the paper.

The paper presents a very interesting complement to previous research made by the same research group. It extends the general validity of the findings made on another protein in another organisms. It is an important contribution to understanding the biology of Asgard archaea and the role played by its cytoskeleton.

I have two major comments, that I divided in sub-comments.

1. Sequence alignments

Parts 1a, 1b and 1e are mandatory, while 1c and 1d are suggestions to increase the breadth of the analysis.

1a. To ensure reproducibility of the sequence alignment parts, I would like the authors to provide more details about how they performed their BLAST search: which type of algorithm, which version of the software, which database, which version of the database, which cut-offs they use for considering a hit as significant, etc.

1b. I am surprised that the protein alignment didn't return more sequences. There are at least 7 MAGs available at NCBI, 5 of which were published before 2018. Were 2DGels found in all of these MAGs? I would like the authors to present the details of the number of sequences found in each group.

1c. I am also curious to hear whether the authors tried to find 2DGels with C-terminal extensions in other Asgard. It would be interesting to see if these extensions are truly limited to Heim and Loki, or if they can be found elsewhere. Also, the authors don't state how frequent the C-term extension is among Lokis. I think these details are useful to assess the universality of the findings of this paper. A phylogenetic tree of the obtained sequences would also give an indication whether the C-terminal extension is the result of convergent evolution and (possibly) horizontal gene transfer, or whether there is a single origin of the acquisition of the extension.

1d. It would also be interesting to know whether the Zn domains (as single-domain proteins) are present beyond Lokis. And, although maybe beyond the scope of this paper, their evolutionary history and whether they can be found in non-Asgard organisms.

1e. With regard to the different affinities for actin of the 2DGels in Loki GC14_75 and MDK1, it would be interesting to know whether there are more than one gelsolin homolog within each of the genomes.

It would make sure that the difference in affinity comes from, as the authors say it, sequence divergence, and not paralogization.

2. Discussion

In general, the discussion is appropriate and the claims are supported by the data.

2a. However, I counted references to two papers only in the whole discussion, which is quite unusual. The authors might want to broaden their scope, discussing their findings in the context of the current literature, when it comes to eukaryogenesis, the potential role of calcium-dependent actin regulation in the little we know of the cytoskeleton of Asgard archaea, etc. A short discussion about calcium metabolism in Asgard might also be interesting, as it seems that Asgard possess a rich collection of calcium channels (<https://doi.org/10.1371/journal.pone.0247806>).

2b. Finally, the authors claim that since Loki-, Heim- and Thorarchaeota show calcium-controlled actin filament disassembly, the last common ancestor (LCA) of Asgard archaea possessed it as well. I believe this is dependent on its presence in other Asgard groups, and on the phylogeny of the various Asgard phyla. Although the latter is not well resolved, there are indications that Asgard might encompass earlier branches (Baldr; <https://doi.org/10.1038/s41564-021-01039-y>). That being said, other sources (<https://doi.org/10.1038/s41586-021-03494-3>) suggest that the presence in these three lineages (excluding HGT) implies that the LCA of Asgard archaea also possesses calcium-controlled actin regulation.

Lionel Guy

Reviewer #3 (Remarks to the Author):

In this manuscript Akil and colleagues report the structural and biochemical characterization of three different gelsolin family members from Asgard archaea. In eukaryotes, Gelsolin family members are Ca²⁺-dependent regulators of the actin cytoskeleton, controlling filament severing and depolymerization. Asgard archaea are the only known prokaryotes to carry homologs for actin and actin regulators, and are considered to be the closest known prokaryotic relatives of eukaryotic cells. Thus, understanding the function and regulation of the asgard actin cytoskeleton is not only important for understanding the physiology of these organisms, but sheds light into the early evolution of eukaryotes as well. As such, these results should be of general interest. The manuscript contains plenty of interesting data, and is written in an easy-to-follow way. However, I think that there are some issues that should be addressed before this article could be published:

Major issues:

- Data for Loki2DGel: At a resolution of 3.25 Å, it seems hard to assign the identity of a metal – the new Ca²⁺ sites in this case – without additional information. At this resolution it would be hard to get right rotameric assignments and backbone atoms aren't really visible. That would make assignment just based on coordination geometry impossible. It is even harder to judge without the actual density being presented. Is there any additional info to confirm that, for instance, the site at the WH2 site is really occupied by a Ca ion?

The data on the function of this protein is also relatively inconclusive. The crystal structure clearly shows an interaction with actin in G-form. Kinetically however, there is a minor effect consistent with filament capping. Pelleting assays show no interaction (or bundling) but then TIRF microscopy shows clear signs of bundling. Any idea what's the source of these differences? The differences are attributed to divergence, for example when compared to MKD1 2DGel. How do these differences look like when mapped on the actin-gelsolin complex? More importantly, how different are the sequences of actin

from Loki GC14_75 and MKD1? As MKD1-2DGel has a strong effect on actin, understanding the sequence differences of the corresponding asgard gelsolin-actin complexes and eukaryotic actin could be key decide whether these differences are indeed due to divergence. From what I can see, the lack of function could be also attributed to the lack of quality of the Loki2DGel prep (Fig S5 for example), which seems to be around 50% pure.

- There are some experiments that were performed on selected gelsolins. Was there a reason for this? For instance, Loki2DGel shows bundling in TIRF but this effect was never show to be Ca sensitive. The far more functional Loki gelsolin from MKD1 is also missing in from the TIRF experiments.

The specific effect of the secondary gelsolin domain of MKD1 got tested independently, but not for any of the other proteins. The secondary domain from Heim2DGel would be arguably the most interesting one, as it shows a different binding mode. Indeed, the paper goes to state that "...However, the actin-bound positions of D2 in the Heim, Loki and Thor G-actin-bound complexes are different , suggesting variation in recycling of actin monomers in these phyla, and indicating that the emergent role of D2 after gene duplication was likely in calcium control of 2DGels, rather than in actin interaction. (page 8)" I am not sure how this would work, does D2 confer Ca regulation beyond what's already present in D1?

- Endogenous asgard actin sequences: Many conclusions make only sense when taking into account the sequence of asgard actins (see above as well). Type I sites are an excellent example. These sites could only be functional if the corresponding residues in actin are conserved in the respective asgard archaea, otherwise the whole claim of Ca²⁺ regulation in these microorganisms would not hold.

- In page 8, the discussion states that "The differential nuclear exclusion of Heim2DGel, in eukaryotic cells (Fig. 6b,c), may hint at a role in directing cellular location." I am not sure how the nuclear exclusion of the protein could point to any compartmentalization within a prokaryote. It sounds to me much more like an artifact due to heterologous expression.

- The introduction states that "however, it is unknown how the force generated from actin polymerization is harnessed for a biological output." I would argue that it is unknown *if* this actin has a cytomotive function. Of course, it is very likely, but very much hypothetical.

- Reproducibility: The methods section states that "Statistics and Reproducibility. All biochemical experiments were repeated 3 times with similar results." Why not simply including the replicates in the paper? As it is it's impossible to judge on the reproducibility of the results.

Minor issues:

- In page 9 the manuscript states: "We speculate that the ancestral Asgard actin cytoskeleton was not highly evolved in the common ancestor..." I am not sure what's meant by this. Perhaps complex would be the right word?

- In page 6 the article states that the 2:1 stoichiometry indicates "...that MKD1-2DGel cooperatively binds to two actin subunits..." I don't think binding cooperativity can be concluded from the data.

- C-terminal domains in X-ray structures: The C-terminal domains of Loki2DGel and Heim2DGel are missing in the crystal structures, so the authors propose they are flexible within the crystal. Is it possible that they have been proteolytically cleaved instead?

- Structure of the C-terminal extensions: The N-terminal end of the linker between the gelsolin domains and the Loki2DGel's zinc finger folds back onto actin. Considering that subsequent regions in the linker are predicted as helical, it is possible that the whole protein stays close to actin. It could be interesting to build an alphafold model of either the whole Loki2DGel, or better yet, its complex with actin.

- The introduction states: "...However, some Asgard profilins do not bind to polyproline (ref 6) motifs while others interact with modest affinities ($\sim 300 \mu\text{M}$)..." This makes it look as if there were more variability than there actually is the function of these proteins. ITC – the technique of choice in ref 6 (also from these authors) – isn't that appropriate for interactions in the $300 \mu\text{M}$ affinity range. All that could be concluded is that at most all profilins have modest affinities for the tested polyprolines.

Reviewer #4 (Remarks to the Author):

This work addresses the fascinating question of the emergence of eukaryotic traits. Specifically, the authors aim to explore the structure and function of the two-domain gelsolins (2DGels), the proteins implicated in remodeling the actin cytoskeleton in Asgard archaea. They express in *E. coli* and purify three different 2DGels: Loki2DGel, MKD1-2DGel, and Heim2DGel. They then test the effects of these proteins on actin polymerization *in vitro* and, in a separate line of experiments, under the conditions of heterologous expression in a model human cell line (U2OS). They conclude that Asgard 2DGels can regulate the polymerization of eukaryotic actin in a calcium-dependent manner. Furthermore, they have determined the X-ray structures of Heim and Loki 2DGels bound to actin monomers and identified the conserved calcium-binding sites required for this function. Based on their data, the authors conclude that the calcium-regulated mechanisms of actin remodeling emerged in the predecessors of the last common ancestor of Asgard archaea. This work supports and further extends the hypothesis put forward by Dr. Robinson's group in their earlier publication on the structure and function of Thor2DGel, a protein from another phylum of Asgard archaea (Akil, et al., 2020, PNAS 117, 19904–19913).

This is high-quality work elucidating a fundamental aspect of eukaryogenesis. However, two issues diminish somewhat my enthusiasm.

The authors present an overarching conclusion emphasizing the similarities and the standard features of the three 2DGels selected for this study. However, despite the sequence similarity and the presence of conserved calcium-binding sites (Fig. 1), the properties of these proteins differ significantly. These differences should be discussed in more detail, not brushed aside. For example, a significant part of the structural analysis is based on the structure of the Loki2DGel-actin complex. In particular, the function of the calcium-binding sites is, in part, derived from this structure. However, Loki2DGel does not depolymerize F-actin. It promotes F-actin bundling, as evidenced by the presented video. In contrast, MKD1-2DGel strongly inhibits actin polymerization, but unfortunately, it did not co-crystallize with actin. This raises the question if the interpretation of the MKD1-2DGel properties based on the Loki2DGel structure is valid. To resolve this conundrum, it might be helpful to analyze in detail the interaction interfaces in the respective models, possibly including the models of the corresponding actins.

The second issue, which does not necessarily reflect negatively on this work, but leaves a certain void in the discussion is the absence of any information on the potential for controlling the free calcium concentration in Asgard archaea. After all, an effective calcium-dependent regulation, as we know it from eukaryotic cells, requires an extensive array of tools such as channels, transporter, sensors, etc. It would be extremely informative to identify their predecessors in archaea.

Reviewers' comments:

Reviewer #1 (Remarks to the Author):

The manuscript is very well written and you can see that the authors have studied the subject extensively. The cross-linking to Norse mythology is very nice. However, the text does not address the effects of pH and temperature on gelsolin and its ability to bind calcium. Nevertheless, the paper has added value for research on gelsolin. Thank you for the positive comments. The suggestion to study the effects of temperature and pH on 2DGel activity is good idea, but out of the scope of the present work. We may pursue this aspect in future work.

Reviewer #2 (Remarks to the Author):

In their contribution, Akil and coauthors show that the gelsolin homologs in two Asgard archaea regulate eukaryotic actin, both in vitro and in eukaryotic cells, and that calcium controls the action of gelsolin. The authors conclude that calcium regulation of actin predates the last common ancestor of Asgard (LCAs).

This review is limited to the protein alignment and searching methods and the evolutionary perspective of the paper.

The paper presents a very interesting complement to previous research made by the same research group. It extends the general validity of the findings made on another protein in another organisms. It is an important contribution to understanding the biology of Asgard archaea and the role played by its cytoskeleton. Thank you for the pointing out the importance of the work and in helping improve the manuscript.

I have two major comments, that I divided in sub-comments.

1. Sequence alignments

Parts 1a, 1b and 1e are mandatory, while 1c and 1d are suggestions to increase the breadth of the analysis.

We have addressed all the comments.

1a. To ensure reproducibility of the sequence alignment parts, I would like the authors to provide more details about how they performed their BLAST search: which type of algorithm, which version of the software, which database, which version of the database, which cut-offs they use for considering a hit as significant, etc.

We now include these details in the methods:

“Default parameters were used in the online blastp interface (<https://blast.ncbi.nlm.nih.gov/>) in searching the non-redundant sequence database (2022/05/21). Potential calcium-binding gelsolin domains were identified by inspection of pairwise sequence alignments to locate the signature “WxG” gelsolin motif and the calcium-binding residues. The searches were not limited by E-value cut-offs.”

1b. I am surprised that the protein alignment didn't return more sequences. There are at least 7 MAGs available at NCBI, 5 of which were published before 2018. Were 2DGels found in all of these MAGs? I would like the authors to present the details of the number of sequences found in each group.

We now clarify the wording to not give the impression that there are just a small number of 2DGels in Asgard. We also reference our earlier publication in which we count the number of 2DGels in 21 MAGs. <https://doi.org/10.1016/j.ceb.2020.08.011> (Table 1).

“BLAST sequence searches revealed that 2DGels exist in all Asgard phyla that are represented in the NCBI non-redundant protein sequences database, with the exception of Odinararchaeota⁷. Thor and Gerdarchaeota MAGs contain 2DGels without C-terminal extensions, whereas Helarchaeota MAGs contain potential “2DGels” that contain a potential divergent gelsolin domain at the N-terminus⁷. Heim MAGs encode a number of hypothetical 2DGels of varied architectures. Examples include, Heim E29_bin46 which...”

1c. I am also curious to hear whether the authors tried to find 2DGels with C-terminal extensions in other Asgard. It would be interesting to see if these extensions are truly limited to Heim and Loki, or if they can be found elsewhere. Also, the authors don't state how frequent the C-term extension is among Lokis. I think these details are useful to assess the universality of the findings of this paper. A phylogenetic tree of the obtained sequences would also give an indication whether the C-terminal extension is the result of convergent evolution and (possibly) horizontal gene transfer, or whether there is a single origin of the acquisition of the extension. As now outlined, in the paragraph above (red), we only found C-terminal extensions in Loki and Heim. We now clarify the frequency of the Loki extension by stating: “The most common 2DGel architecture, among Loki species,....”

We have chosen not to include a phylogenetic tree, although we agree that it is an interesting exercise. We may pursue this in future work.

1d. It would also be interesting to know whether the Zn domains (as single-domain proteins) are present beyond Lokis. And, although maybe beyond the scope of this paper, their evolutionary history and whether they can be found in non-Asgard organisms.

The Zn domains are mainly, if not exclusively in Lokis. There are a few metagenomics sequences that are not assigned to Loki, such as KAF5435067.1. However, we have low confidence in these reports, since they have not been sequenced/assembled more than once, and this sequence shares 93% identity with MKD1. We now comment:

“We were not able to consistently detect homologous Zn domains in other organisms beyond Loki.”

1e. With regard to the different affinities for actin of the 2DGels in Loki GC14_75 and MDK1, it would be interesting to know whether there are more than one gelsolin homolog within each of the genomes. It would make sure that the difference in affinity comes from, as the authors say it, sequence divergence, and not paralogization.

This is an important point, we now comment in the discussion:

“Inspection of the numbers of gelsolin-like proteins in both species of Loki is revealing. The fully sequenced MKD1 genome has a single 2DGel, two ProGels and two 1DGelXs. By contrast, the incomplete GC14_75 MAG has two 2DGels (Supplementary Fig. 1a), four ProGels and three 1DGelXs. Thus, the differences in sequence that result differences in the activities of Loki and MKD1 2DGels may represent functional diversification through paralogization in these species, and/or an *in vitro* variance in modulating the non-native rActin.”

2. Discussion

In general, the discussion is appropriate and the claims are supported by the data.
Thank you for this comment.

2a. However, I counted references to two papers only in the whole discussion, which is quite unusual. The authors might want to broaden their scope, discussing their findings in the context of the current literature, when it comes to eukaryogenesis, the potential role of calcium-dependent actin regulation in the little we know of the cytoskeleton of Asgard archaea, etc. A short discussion about calcium metabolism in Asgard might also be interesting, as it seems that Asgard possess a rich collection of calcium channels (<https://doi.org/10.1371/journal.pone.0247806>).

Thank you for pointing out this recent and very relevant publication. We have incorporated it into the discussion:

“Sequence analysis has identified several potential eukaryotic-like calcium channels in Asgard archaea genomes, which have been predicted to control calcium influx into these organisms²⁹. In particular, Loki, Heim and Thor contain potential Calcium Load-activation Calcium Channels (CLAC), and Loki and Heim potential presenilins²⁹. Thus, Asgard archaea may use calcium channels to regulate calcium levels, and in turn, control actin dynamics.”

2b. Finally, the authors claim that since Loki-, Heim- and Thorarchaeota show calcium-controlled actin filament disassembly, the last common ancestor (LCA) of Asgard archaea possessed it as well. I believe this is dependent on its presence in other Asgard groups, and on the phylogeny of the various Asgard phyla. Although the latter is not well resolved, there are indications that Asgard might encompass earlier branches (Baldr; <https://doi.org/10.1038/s41564-021-01039-y>). That being said, other sources (<https://doi.org/10.1038/s41586-021-03494-3>) suggest that the presence in these three lineages (excluding HGT) implies that the LCA of Asgard archaea also possesses calcium-controlled actin regulation.

This is a good reminder that the phylogeny is still open to question. We have moderated the discussion:

“In summary, calcium-controlled actin filament disassembly is present within the 2DGel proteins from Loki, Heim and Thor Asgard phyla. This indicates that their last common ancestor will have possessed a 2DGel which responded to calcium ions.”

and

“Nonetheless, the last common ancestor of Loki, Heim and Thor, and possibly the last Asgard archaea common ancestor, will have possessed an actin cytoskeleton under the control of calcium signalling, and these features predate eukaryogenesis.”

And in the abstract:

“We conclude that the calcium-regulated actin cytoskeleton predates eukaryogenesis and emerged in the predecessors of the last common ancestor of Loki, Heim and Thorarchaeota.”

And in the intro:

“This indicates that the last Loki, Heim and Thor common ancestor likely possessed a calcium-regulated 2DGel actin-filament depolymerization system, and that the emergence of calcium regulation of actin dynamics predates eukaryogenesis.”

Lionel Guy

Reviewer #3 (Remarks to the Author):

In this manuscript Akil and colleagues report the structural and biochemical characterization of three different gelsolin family members from Asgard archaea. In eukaryotes, Gelsolin family members are Ca²⁺-dependent regulators of the actin

cytoskeleton, controlling filament severing and depolymerization. Asgard archaea are the only known prokaryotes to carry homologs for actin and actin regulators, and are considered to be the closest known prokaryotic relatives of eukaryotic cells. Thus, understanding the function and regulation of the asgard actin cytoskeleton is not only important for understanding the physiology of these organisms, but sheds light into the early evolution of eukaryotes as well. As such, these results should be of general interest. The manuscript contains plenty of interesting data, and is written in an easy-to-follow way. However, I think that there are some issues that should be addressed before this article could be published:

Thank you for the nice synopsis and enthusiasm for the work.

Major issues:

- Data for Loki2DGel: At a resolution of 3.25 Å, it seems hard to assign the identity of a metal – the new Ca²⁺ sites in this case – without additional information. At this resolution it would be hard to get right rotameric assignments and backbone atoms aren't really visible. That would make assignment just based on coordination geometry impossible. It is even harder to judge without the actual density being presented. Is there any additional info to confirm that, for instance, the site at the WH2 site is really occupied by a Ca ion?

We agree with the reviewer that the placement of metal ions, at this resolution, should be very carefully conducted. However, we are confident that assignment of the cation as calcium is correct. To justify the placement of the calcium ion we have added Fig. 2b, which includes the Fo-Fc map and bond distances. This clearly indicates that a cation occupies the site. In the text, we now explain our procedure for identifying this ion as calcium.

"The Fo-Fc electron density map showed an 8.5 σ peak at this site, prior to the addition of the cation to the structure (Fig. 2b). The bonding distances of the coordination sphere atoms (2.23-2.40 Å) are appropriate for several types of cation from the crystallization conditions, Ca²⁺, Mg²⁺ or Na⁺. We refined each of these cations individually in the structure. Only, Ca²⁺ refined normally, characterized by a B-factor similar to the coordination sphere atoms and no unexplained Fo-Fc electron density (>3.0 σ)."

The data on the function of this protein is also relatively inconclusive. The crystal structure clearly shows an interaction with actin in G-form. Kinetically however, there is a minor effect consistent with filament capping. Pelleting assays show no interaction (or bundling) but then TIRF microscopy shows clear signs of bundling. Any idea what's the source of these differences?

We attribute the differences to very weak interactions. At the high concentrations and stoichiometric ratio in the crystallization conditions, the balance is shifted towards interaction. In the other assays, actin:actin interactions dominate over actin:Loki2DGel interactions. The pelleting assay and capping are consistent, since the low number of filament ends would not reveal capping proteins in the sedimentation assay. The molecular crowding conditions, within the TIRF assay, will tend to bundle filaments due to the excluded volume effect. We now show that the bundling in the presence of Loki2DGel is calcium dependent (Fig. 5e), demonstrating that it protein dependent. We suggest that the correct GC14_75 actin would be needed to ascertain whether these activities are the entire functionality of the protein, or whether it may act more like MKD1-2DGel. Unfortunately, we are not able to produce suitable amounts of functional Asgard actins. We include a section in the discussion detailing the numbers of gelsolins in each species:

“Inspection of the numbers of gelsolin-like proteins in both species of Loki is revealing. The fully sequenced MKD1 genome has a single 2DGel, two ProGels and two 1DGelXs. By contrast, the incomplete GC14_75 MAG has two 2DGels (Supplementary Fig. 1a), four ProGels and three 1DGelXs. Thus, the differences in sequence that result differences in the activities of Loki and MKD1 2DGels may represent functional diversification through paralogization in these species, and/or an *in vitro* variance in modulating the non-native rActin.”

The differences are attributed to divergence, for example when compared to MKD1 2DGel. How do these differences look like when mapped on the actin-gelsolin complex?

We now map the interaction residues onto the sequence alignment in Supplementary Fig. 1a.

More importantly, how different are the sequences of actin from Loki GC14_75 and MKD1? As MKD1-2DGel has a strong effect on actin, understanding the sequence differences of the corresponding asgard gelsolin-actin complexes and eukaryotic actin could be key decide whether these differences are indeed due to divergence. We map the interaction residues onto a sequence alignment in Supplementary Fig. 1c.

We comment on these last two points – in the results:

“The actin-binding residues on Loki2DGel show a core conservation with MKD1-2DGel with a few residues of opposite character (Supplementary Fig. 1a). The actin sequences from Loki GC14_75 and MKD1 are identical in the Loki2DGel-binding site, and these residues show remarkable conservation with rActin (Supplementary Fig. 1c), in line with the complex formation between Asgard 2DGels and rActin⁵.”

And in the discussion:

“Interestingly, MKD1-2DGel has a two-residue insertion at the end of the D1 long helix (Fig. 1b), which will increase the steric clash with the lower actin (Fig. 7e, arrow), possibly providing an explanation for its superior filament severing activity.”

And in Supplementary Fig. 1 legend:

“Asterisks above the alignment signify residues that contact the actin subunits in the Loki2DGel complex, lower actin (green) and upper actin (brown). Insertions and substitutions between Loki and MKD1 sequences, such as F72D and V99K, likely provide the basis for the differences in activity.”

From what I can see, the lack of function could be also attributed to the lack of quality of the Loki2DGel prep (Fig S5 for example), which seems to be around 50% pure.

The protein purity in the previous Fig S5 was not representative. We have now replaced that figure. The higher purity preps do not show an increased functionality. We do not think that protein quality is the basis for the difference in activity.

- *There are some experiments that were performed on selected gelsolins. Was there a reason for this? For instance, Loki2DGel shows bundling in TIRF but this effect was never show to be Ca sensitive. The far more functional Loki gelsolin from MKD1 is also missing in from the TIRF experiments.*

We have rectified this oversight. The Loki2DGel calcium/EGTA matching conditions are now shown in Fig. 5e and in Movie 1. The MKD1-2DGel calcium/EGTA matching conditions are shown in Fig. 5f,g and Movies 2 and 3.

The specific effect of the secondary gelsolin domain of MKD1 got tested independently, but not for any of the other proteins. The secondary domain from

Heim2DGel would be arguably the most interesting one, as it shows a different binding mode.

We tested the MKD1-D2 construct since we do not have a crystal structure of the MKD1/actin complex. This demonstrated that D2 binds to actin, as would be expected from the Loki2DGel structure. Further dissection of the activities of the domains of the 2DGels, including Heim2DGel, are experiments that we may pursue in future work.

Indeed, the paper goes to state that “..However, the actin-bound positions of D2 in the Heim, Loki and Thor G-actin-bound complexes are different , suggesting variation in recycling of actin monomers in these phyla, and indicating that the emergent role of D2 after gene duplication was likely in calcium control of 2DGels, rather than in actin interaction. (page 8)” I am not sure how this would work, does D2 confer Ca regulation beyond what’s already present in D1?

We have removed this point about D2 regulation by Ca. We were suggesting that an emergent role of D2 after gene duplication was likely in calcium control of 2DGels, by obscuring the actin-binding sites on D1 in the calcium-free conformation. However, we agree that this is too speculative.

- Endogenous asgard actin sequences: Many conclusions make only sense when taking into account the sequence of asgard actins (see above as well). Type I sites are an excellent example. These sites could only be functional if the corresponding residues in actin are conserved in the respective asgard archaea, otherwise the whole claim of Ca²⁺ regulation in these microorganisms would not hold.

The actin sequence alignment now appears in Supplementary Fig. 1c. The Type I site (Glu167) is conserved. We comment at the relevant places on the conservation, such as:

“The Type I site in D1 coordinates rActin residue Glu167, which is conserved in Loki and MKD1 actins...”

- In page 8, the discussion states that “The differential nuclear exclusion of Heim2DGel, in eukaryotic cells (Fig. 6b,c), may hint at a role in directing cellular location.” I am not sure how the nuclear exclusion of the protein could point to any compartmentalization within a prokaryote. It sounds to me much more like an artifact due to heterologous expression.

We are not suggesting compartmentalization. We agree that it may also be due to heterologous expression. The use of the words “may hint” clearly highlight this point as speculation, which should be allowed in the discussion.

*- The introduction states that “however, it is unknown how the force generated from actin polymerization is harnessed for a biological output.” I would argue that it is unknown *if* this actin has a cytomotive function. Of course, it is very likely, but very much hypothetical.*

Changed to:

“it is unknown if and how the force generated from actin polymerization is harnessed for a biological output.”

- Reproducibility: The methods section states that “Statistics and Reproducibility. All biochemical experiments were repeated 3 times with similar results.” Why not simply including the replicates in the paper? As it is it’s impossible to judge on the reproducibility of the results.

The assembly and disassembly replicates now appear in Supplementary Fig. 11.

Minor issues:

- In page 9 the manuscript states: “We speculate that the ancestral Asgard actin cytoskeleton was not highly evolved in the common ancestor...” I am not sure what’s meant by this. Perhaps complex would be the right word?

Changed to “immature”.

- In page 6 the article states that the 2:1 stoichiometry indicates “...that MKD1-2DGel cooperatively binds to two actin subunits...” I don’t think binding cooperativity can be concluded from the data.

Changed to:

“indicating that MKD1-2DGel may bind cooperatively to two actin subunits”

- C-terminal domains in X-ray structures: The C-terminal domains of Loki2DGel and Heim2DGel are missing in the crystal structures, so the authors propose they are flexible within the crystal. Is it possible that they have been proteolytically cleaved instead?

We now state in the methods:

“The lack of density for the C-terminal extension of Heim2DGel and Loki2DGel may be due to flexibility in these regions or to proteolytic cleavage in the crystallization drops.”

- Structure of the C-terminal extensions: The N-terminal end of the linker between the gelsolin domains and the Loki2DGel’s zinc finger folds back onto actin. Considering that subsequent regions in the linker are predicted as helical, it is possible that the whole protein stays close to actin. It could be interesting to build an alphafold model of either the whole Loki2DGel, or better yet, its complex with actin. The actin-bound models now appear in Supplementary Fig. 10. The Tails do not form meaningful interactions. We calculated the AF2 models of the 2DGel proteins in isolation, again there was no meaningful interactions with the tails (these models are not included in the revised manuscript). We have updated the discussion to reflect the extra information in the actin-bound models.

“For comparison, we calculated AlphaFold2 (AF2) predicted structures of Loki, Heim, MKD1 and Thor 2DGels bound to their respective Asgard actin (Supplementary Fig. 10). The lower actins, in each model, adopt an F-actin conformation and each D2 is located to the side of the two-actin modelled filament (Supplementary Fig. 10). We interpret these AF2 models to indicate the initial contact with actin filaments, where D2 and the WH2-like motif locate the 2DGels to the side of a filament allowing competition between D1 and the lower actin to induce severing (Fig. 7d,e). Interestingly, MKD1-2DGel has a two-residue insertion at the end of the D1 long helix (Fig. 1b), which will increase the steric clash with the lower actin (Fig. 7e, arrow), possibly providing an explanation for its superior filament severing activity.”

- The introduction states: “...However, some Asgard profilins do not bind to polyproline (ref 6) motifs while others interact with modest affinities (~300 μ M)...” This makes it look as if there were more variability than there actually is the function of these proteins. ITC – the technique of choice in ref 6 (also from these authors) – isn’t that appropriate for interactions in the 300 μ M affinity range. All that could be concluded is that at most all profilins have modest affinities for the tested polyprolines.

Here, we are quoting the numbers from the literature. We have changed the sentence with regards to eukaryotes to highlight that point:

“In eukaryotes, more robust profilin/polyproline motif interactions (10-100 μM)^{10,11} have been reported to recruit profilin:actin to filament nucleation machineries”

Reviewer #4 (Remarks to the Author):

This work addresses the fascinating question of the emergence of eukaryotic traits. Specifically, the authors aim to explore the structure and function of the two-domain gelsolins (2DGels), the proteins implicated in remodeling the actin cytoskeleton in Asgard archaea. They express in E. coli and purify three different 2DGels: Loki2DGel, MKD1-2DGel, and Heim2DGel. They then test the effects of these proteins on actin polymerization in vitro and, in a separate line of experiments, under the conditions of heterologous expression in a model human cell line (U2OS). They conclude that Asgard 2DGels can regulate the polymerization of eukaryotic actin in a calcium-dependent manner. Furthermore, they have determined the X-ray structures of Heim and Loki 2DGels bound to actin monomers and identified the conserved calcium-binding sites required for this function. Based on their data, the authors conclude that the calcium-regulated mechanisms of actin remodeling emerged in the predecessors

of the last common ancestor of Asgard archaea. This work supports and further extends the hypothesis put forward by Dr. Robinson's group in their earlier publication on the structure and function of Thor2DGel, a protein from another phylum of Asgard archaea (Akil, et al., 2020, PNAS 117, 19904–19913).

This is high-quality work elucidating a fundamental aspect of eukaryogenesis. Thank you for the positive statements and clear summary of the manuscript.

However, two issues diminish somewhat my enthusiasm.

The authors present an overarching conclusion emphasizing the similarities and the standard features of the three 2DGels selected for this study. However, despite the sequence similarity and the presence of conserved calcium-binding sites (Fig. 1), the properties of these proteins differ significantly. These differences should be discussed in more detail, not brushed aside. For example, a significant part of the structural analysis is based on the structure of the Loki2DGel-actin complex. In particular, the function of the calcium-binding sites is, in part, derived from this structure. However, Loki2DGel does not depolymerize F-actin. It promotes F-actin bundling, as evidenced by the presented video. In contrast, MKD1-2DGel strongly inhibits actin polymerization, but unfortunately, it did not co-crystallize with actin. This raises the question if the interpretation of the MKD1-2DGel properties based on the Loki2DGel structure is valid. To resolve this conundrum, it might be helpful to analyze in detail the interaction interfaces in the respective models, possibly including the models of the corresponding actins.

A similar comment was raised by reviewer 2. To address this we have mapped the actin interaction residues onto the 2DGel sequence alignment in Supplementary Fig. 1a, and have also mapped the Loki2DGel interaction site onto a sequence alignment of the two respective actins (Supplementary Fig. 1c).

These figures show that: 1) the Loki and MKD1 actins have identical residues in the Loki2DGel binding site; 2) the Loki and MKD1 2DGels have a conserved core of

residues in their actin binding sites; and 3) all of the calcium-binding sites are conserved between the Loki and MKD1 2DGels. Together, this conservation in binding sites justifies the structural comparison.

We comment – in the results:

“The actin-binding residues on Loki2DGel show a core conservation with MKD1-2DGel with a few residues of opposite character (Supplementary Fig. 1a). The actin sequences from Loki GC14_75 and MKD1 are identical in the Loki2DGel-binding site, and these residues show remarkable conservation with rActin (Supplementary Fig. 1c), in line with the complex formation between Asgard 2DGels and rActin⁵.”

And in the discussion:

“Interestingly, MKD1-2DGel has a two-residue insertion at the end of the D1 long helix (Fig. 1b), which will increase the steric clash with the lower actin (Fig. 7e, arrow), possibly providing an explanation for its superior filament severing activity.”

And:

“Inspection of the numbers of gelsolin-like proteins in both species of Loki is revealing. The fully sequenced MKD1 genome has a single 2DGel, two ProGels and two 1DGelXs. By contrast, the incomplete GC14_75 MAG has two 2DGels (Supplementary Fig. 1a), four ProGels and three 1DGelXs. Thus, the differences in sequence that result differences in the activities of Loki and MKD1 2DGels may represent functional diversification through paralogization in these species, and/or an *in vitro* variance in modulating the non-native rActin.”

And in Supplementary Fig. 1 legend:

“Asterisks above the alignment signify residues that contact the actin subunits in the Loki2DGel complex, lower actin (green) and upper actin (brown). Insertions and substitutions between Loki and MKD1 sequences, such as F72D and V99K, likely provide the basis for the differences in activity.”

We include the AF2 models of Loki2DGel and MKD1-2DGel bound to actin (Supplementary Fig. 10). And comment in the discussion:

“AF2 predicted structures of Loki and MKD1 2DGels bound to their respective Asgard actins also support a similar binding mode for Loki and MKD1 2DGels (Supplementary Fig. 10).”

The second issue, which does not necessarily reflect negatively on this work, but leaves a certain void in the discussion is the absence of any information on the potential for controlling the free calcium concentration in Asgard archaea. After all, an effective calcium-dependent regulation, as we know it from eukaryotic cells, requires an extensive array of tools such as channels, transporter, sensors, etc. It would be extremely informative to identify their predecessors in archaea.

We now include a short discussion relevant to calcium channels:

“Sequence analysis has identified several potential eukaryotic-like calcium channels in Asgard archaea genomes, which have been predicted to control calcium influx into these organisms²⁹. In particular, Loki, Heim and Thor contain potential Calcium Load-activation Calcium Channels (CLAC), and Loki and Heim potential presenilins²⁹. Thus, Asgard archaea may use calcium channels to regulate calcium levels, and in turn, control actin dynamics.”